# ViTPose: Simple Vision Transformer Baselines for Human Pose Estimation

**Yufei Xu**[1][*],   **Jing Zhang**[1][*],   **Qiming Zhang**[1],   **Dacheng Tao**[2,1]

[1]School of Computer Science, The University of Sydney, Australia

[2]JD Explore Academy, China

`yuxu7116@uni.sydney.edu.au, jing.zhang1@sydney.edu.au,`
`qzha2506@uni.sydney.edu.au, dacheng.tao@gmail.com`

## Abstract

Although no specific domain knowledge is considered in the design, plain vision transformers have shown excellent performance in visual recognition tasks. However, little effort has been made to reveal the potential of such simple structures for pose estimation tasks. In this paper, we show the surprisingly good capabilities of plain vision transformers for pose estimation from various aspects, namely simplicity in model structure, scalability in model size, flexibility in training paradigm, and transferability of knowledge between models, through a simple baseline model called **ViTPose**. Specifically, ViTPose employs plain and non-hierarchical vision transformers as backbones to extract features for a given person instance and a lightweight decoder for pose estimation. It can be scaled up from 100M to 1B parameters by taking the advantages of the scalable model capacity and high parallelism of transformers, setting a new Pareto front between throughput and performance. Besides, ViTPose is very flexible regarding the attention type, input resolution, pre-training and finetuning strategy, as well as dealing with multiple pose tasks. We also empirically demonstrate that the knowledge of large ViTPose models can be easily transferred to small ones via a simple knowledge token. Experimental results show that our basic ViTPose model outperforms representative methods on the challenging MS COCO Keypoint Detection benchmark, while the largest model sets a new state-of-the-art, i.e., 80.9 AP on the MS COCO test-dev set. The code and models are available at https://github.com/ViTAE-Transformer/ViTPose.

## 1   Introduction

Human pose estimation is one of the fundamental tasks in computer vision and has a wide range of real-world applications [51, 29]. It aims to localize human anatomical keypoints and is challenging due to the variations of occlusion, truncation, scales, and human appearances. To deal with these issues, there has been rapid progress in deep learning-based methods [37, 42, 36, 50], which typically tackle the challenging task using convolutional neural networks.

Recently, vision transformers [13, 31, 10, 34, 32] have shown great potential in many vision tasks. Inspired by their success, different vision transformer structures have been deployed for the pose estimation task. Most of them adopt a CNN as a backbone and then use a transformer of elaborate structures to refine the extracted features and model the relationship between the body keypoints. For example, PRTR [23] incorporates both transformer encoders and decoders to gradually refine the locations of the estimated keypoints in a cascade manner. TokenPose [27] and TransPose [44], instead, adopt an encoder-only transformer structure to process the features extracted by CNNs. On

---

[*]Equal contribution.

the other hand, HRFormer [48] employs the transformer to directly extract features and introduce high-resolution representations via multi-resolution parallel transformer modules. These methods have obtained superior performance on pose estimation tasks. However, they either need extra CNNs for feature extraction or require careful designs of the transformer structure to adapt to the task. This motivates us to think from an opposite direction, *how well can the plain vision transformer do for pose estimation?*

To find the answer to this question, we propose a simple baseline model called **ViTPose** and demonstrate its potential on the MS COCO Keypoint dataset [28]. Specifically, ViTPose employs plain and non-hierarchical vision transformers [13] as backbones to extract feature maps for the given person instances, where the backbones are pre-trained with masked image modeling pretext tasks, *e.g.*, MAE [15], to provide a good initialization. Then, a following lightweight decoder processes the extracted features by upsampling the feature maps and regressing the heatmaps w.r.t. the keypoints, which is composed of two deconvolution layers and one prediction layer. Despite no elaborate designs in the model, ViTPose obtains state-of-the-art (SOTA) performance of 80.9 AP on the challenging MS COCO Keypoint test-dev set. It should be noted that this paper does not claim the algorithmic superiority but rather presents a simple and solid transformer baseline with superior performance for pose estimation.

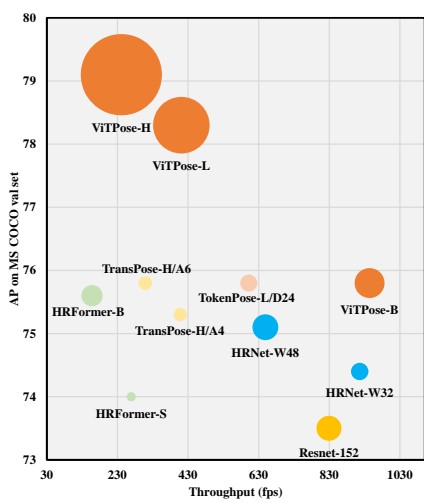

Figure 1: The comparison of ViTPose and SOTA methods on MS COCO val set regarding model size, throughput, and precision. The size of each bubble represents the number of model parameters.

Besides the superior performance, we also show the surprisingly good capabilities of ViTPose from various aspects, namely simplicity, scalability, flexibility, and transferability. 1) For simplicity, thanks to vision transformers' strong feature representation ability, the ViTPose framework can be extremely simple. For example, it does not require any specific domain knowledge for the design of the backbone encoder and enjoys a plain and non-hierarchical encoder structure by simply stacking several transformer layers. The decoder can be further simplified to a single up-sampling layer followed by a convolutional prediction layer with a negligible performance drop. Such a structural simplicity makes ViTPose enjoy better parallelism so that it reaches a new Pareto front in terms of the inference speed and performance, as shown in Fig. 1. 2) In addition, the simplicity in structure brings the excellent scalability properties of ViTPose. Thus it benefits from the rapid development of scalable pre-trained vision transformers. Specifically, one can easily control the model size by stacking different numbers of transformer layers and increasing or decreasing the feature dimensions, *e.g.*, using ViT-B, ViT-L, or ViT-H, to balance the inference speed and performance for various deployment requirements. 3) Furthermore, we demonstrate that ViTPose is very flexible in the training paradigm. ViTPose can adapt well to different input resolutions and feature resolutions with minor modifications and can invariably deliver more accurate pose estimation results for higher resolution inputs. Apart from training the ViTPose on a single pose dataset as the common practice, we can modify it to adapt to multiple pose datasets by adding extra decoders very flexibly, resulting in a joint training pipeline and bringing significant performance improvement. This training paradigm brings only marginal (extra) computational cost since the decoder in ViTPose is rather lightweight. In addition, ViTPose can still obtain SOTA performance when pre-trained using smaller unlabelled datasets or finetuned with the attention modules frozen, requiring less training cost than a fully pre-trained finetuning paradigm. 4) Last but not least, the performance of small ViTPose models can be easily improved by transferring the knowledge from large ViTPose models through an extra learnable knowledge token, demonstrating a good transferability of ViTPose.

In conclusion, the contribution of this paper is threefold. 1) We propose a simple yet effective baseline model named ViTPose for human pose estimation. It obtains SOTA performance on the MS COCO Keypoint dataset even without the usage of elaborate structural designs or complex frameworks. 2)

The simple ViTPose model demonstrates to have surprisingly good capabilities, including structural simplicity, model size scalability, training paradigm flexibility, and knowledge transferability. These capabilities build a strong baseline for vision transformer-based pose estimation tasks and would possibly shed light on further development in the field. 3) Comprehensive experiments on popular benchmarks are conducted to study and analyze the capabilities of ViTPose. With a very big vision transformer model as the backbone, *i.e.*, ViTAE-G [52], a single ViTPose model obtains the best 80.9 AP on the MS COCO Keypoint test-dev set.

## 2 Related Work

### 2.1 Vision transformer for pose estimation

Pose estimation has experienced rapid development from CNNs [42] to vision transformer networks. Early works tend to treat transformer as a better decoder [23, 27, 44], *e.g.*, TransPose [44] directly processes the features extracted by CNNs to model the global relationship. TokenPose [27] proposes token-based representations by introducing extra tokens to estimate the locations of occluded keypoints and model the relationship among different keypoints. To get rid of the CNNs for feature extraction, HRFormer [48] is proposed to use transformers to extract high-resolution features directly. A delicate parallel transformer module is proposed to fuse multi-resolution features in HRFormer gradually. These transformer-based pose estimation methods obtain superior performance on popular keypoint estimation benchmarks. However, they either need CNNs for feature extraction or require careful designs of the transformer structures. There have been little efforts in exploring the potential of plain vision transformers for the pose estimation tasks. In this paper, we fill this gap by proposing a simple yet effective baseline model, ViTPose, based on the plain vision transformers.

### 2.2 Vision transformer pre-training

Inspired by the success of ViT [13], many different vision transformer backbones [31, 43, 40, 55, 39, 52, 38, 53] have been proposed, which are typically trained on the ImageNet-1K [12] dataset in a fully supervised setting. Recently, self-supervised learning methods [15, 4] have been proposed for training plain vision transformers. With masked image modeling (MIM) as pretext tasks, these methods provide good initializations for plain vision transformers. In this paper, we focus on the pose estimation tasks and adopt plain vision transformers with MIM pre-training as backbones. Besides, we explore whether pre-training using ImageNet-1K is necessary for pose estimation tasks. Surprisingly, we find that pre-training using smaller unlabelled pose datasets can also provide a good initialization for the pose estimation tasks.

## 3 ViTPose

### 3.1 The simplicity of ViTPose

**Structure simplicity.** The goal of this paper is to provide a simple yet effective vision transformer baseline for pose estimation tasks and explore the potential of plain and non-hierarchical vision transformers [13]. Thus, we keep the structure as simple as possible and try to avoid fancy but complex modules, even though they may improve performance. To this end, we simply append several decoder layers after the transformer backbone to estimate the heatmaps w.r.t. the keypoints, as shown in Fig. 2 (a). For simplicity, we do not adopt skip-connections or cross-attentions in the decoder layers but simple deconvolution layers and a prediction layer, as in [42]. Specifically, given a person instance image $X \in \mathcal{R}^{H \times W \times 3}$ as input, ViTPose first embeds the images into tokens via a patch embedding layer, *i.e.*, $F \in \mathcal{R}^{\frac{H}{d} \times \frac{W}{d} \times C}$, where $d$ (*e.g.*, 16 by default) is the downsampling ratio of the patch embedding layer, and $C$ is the channel dimension. After that, the embedded tokens are processed by several transformer layers, each of which is consisted of a multi-head self-attention (MHSA) layer and a feed-forward network (FFN), *i.e.*,

$$F_{i+1}^{'} = F_i + \text{MHSA}(\text{LN}(F_i)), \quad F_{i+1} = F_{i+1}^{'} + \text{FFN}(\text{LN}(F_{i+1}^{'})), \tag{1}$$

where $i$ represents the output of the $i$th transformer layer and the initial feature $F_0 = \text{PatchEmbed}(X)$ denotes the features after the patch embedding layer. It should be noted that

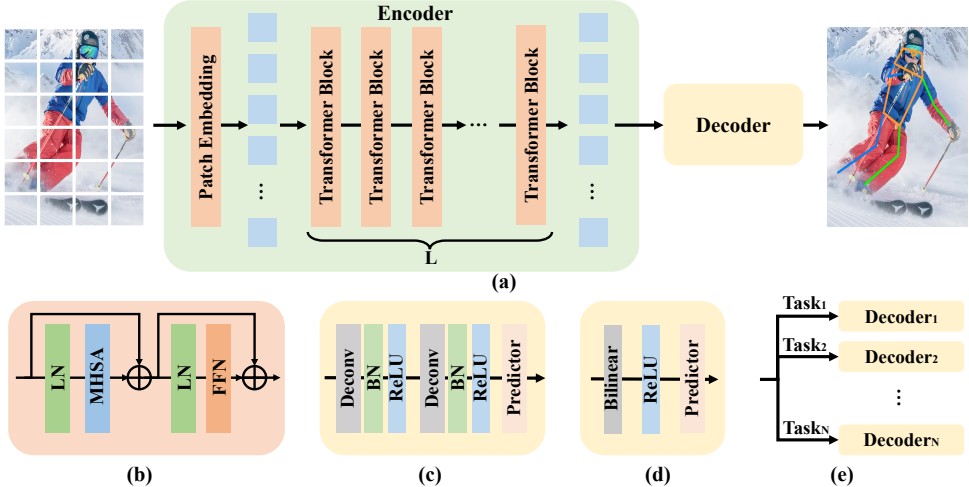

Figure 2: (a) The framework of ViTPose. (b) The transformer block. (c) The classic decoder. (d) The simple decoder. (e) The decoders for multiple datasets.

the spatial and channel dimensions are constant for each transformer layer. We denote the output feature of the backbone network as $F_{out} \in \mathcal{R}^{\frac{H}{d} \times \frac{W}{d} \times C}$.

We adopt two kinds of lightweight decoders to process the features extracted from the backbone network and localize the keypoints. The first one is the classic decoder. It is composed of two deconvolution blocks, each of which contains one deconvolution layer followed by batch normalization [19] and ReLU [1]. Following the common setting of previous methods [42, 50], each block upsamples the feature maps by 2 times. Then, a convolution layer with the kernel size $1 \times 1$ is utilized to get the localization heatmaps for the keypoints, *i.e.*,

$$K = \text{Conv}_{1\times1}(\text{Deconv}(\text{Deconv}(F_{out}))), \tag{2}$$

where $K \in \mathcal{R}^{\frac{H}{4} \times \frac{W}{4} \times N_k}$ denotes the estimated heatmaps (one for each keypoint) and $N_k$ is the number of keypoints to be estimated, which is set to 17 for the MS COCO dataset.

Although the classic decoder is simple and lightweight, we also try another simpler decoder in ViTPose, which is proved effective thanks to the strong representation ability of the vision transformer backbone. Specifically, we directly upsample the feature maps by 4 times with bilinear interpolation, followed by a ReLU and a convolution layer with the kernel size $3 \times 3$ to get the heatmaps, *i.e.*,

$$K = \text{Conv}_{3\times3}(\text{Bilinear}(\text{ReLU}(F_{out}))). \tag{3}$$

Despite the less non-linear capacity of this simpler decoder, it obtains competitive performance compared with the classic one and the carefully designed transformer-based decoders in previous representative methods, demonstrating the structure simplicity of ViTPose.

### 3.2 The scalability of ViTPose

Since ViTPose enjoys the structure simplicity, one can pick a point at the new Pareto front in Fig. 1 according to the deployment requirements and easily control the model size accordingly by stacking different numbers of transformer layers and increasing or decreasing the feature dimensions. In this sense, ViTPose can benefit from the rapid development of scalable pre-trained vision transformers without much modifications to the other parts. To investigate the scalability of ViTPose, we use the pre-trained backbones of different model capacities and finetune them on the MS COCO dataset. For example, we use ViT-B, ViT-L, ViT-H [13], and ViTAE-G [52] with the classic decoder for pose estimation and observe consistent performance gains with the model size increasing. For ViT-H and ViTAE-G, which use patch embedding with size $14 \times 14$ during pre-training, we use zero padding to formulate a patch embedding with size $16 \times 16$ for the same setting with ViT-B and ViT-L.

### 3.3 The flexibility of ViTPose

**Pre-training data flexibility.** ImageNet [12] pre-training of the backbone networks has been a *de facto* routine for a good initialization. However, it requires extra data beyond the pose ones, which makes the data requirement higher for the pose estimation task. It comes to us whether we can use only the pose data during the whole training phase to relax the data requirement. To explore the data flexibility, apart from the default settings of ImageNet [12] pre-training, we use MAE [15] to pre-train the backbones with MS COCO [28] and a combination of MS COCO and AI Challenger [41] respectively by random masking 75% patches from the images and reconstructing those masked patches. Then, we use the pre-trained weights to initialize the backbones of ViTPose and finetune the model on the MS COCO dataset. Surprisingly, although the volume of the pose data is much smaller than ImageNet, ViTPose trained with pose data only can obtain competitive performance, implying that ViTPose can learn a good initialization flexibly from data of different scales.

**Resolution flexibility.** We vary the input image size and downsampling ratios $d$ of ViTPose to evaluate its flexibility regarding the input and feature resolution. Specifically, to adapt ViTPose to input images at higher resolutions, we simply resize the input images and train the model on them accordingly. Besides, to adapt the model to lower downsampling ratios, *i.e.*, higher feature resolutions, we simply change the stride of the patch embedding layer to partition tokens with overlap and retain the size of each patch. We show that the performance of ViTPose increases consistently regarding either higher input resolution or higher feature resolution.

**Attention type flexibility.** Using full attention on higher resolution feature maps will cause a huge memory footprint and computational cost due to the quadratic computational complexity and memory consumption of attention calculation. Window-based attention with relative position embedding [25, 26] has been explored to alleviate the heavy memory burden of dealing with the higher resolution feature maps. However, simply using window-based attention for all transformer blocks degrades the performance due to the lack of global context modeling ability. To address the problem, we adopt two techniques, *i.e.*, 1) *Shift window:* Instead of using fixed windows for attention calculation, we use shift-window mechanism [31] to help broadcast the information between adjacent windows; and 2) *Pooling window.* Apart from the shift window mechanism, we try another solution via pooling. Specifically, we pool the tokens for each window to get the global context feature within the window. These features are then fed into each window to serve as key and value tokens to enable cross-window feature communication. Besides, we prove that the two strategies are complementary to each other and can work together to improve the performance and reduce memory footprint, without the need of extra parameters or modules but with simple modifications to the attention calculation.

**Finetuning flexibility.** As demonstrated in NLP fields [30, 2], pre-trained transformer models can well generalize to other tasks with partial parameters tuning. To investigate whether it still holds for vision transformers, we finetune ViTPose on MS COCO with all parameters unfrozen, MHSA modules frozen, and FFN modules frozen, respectively. We empirically demonstrate that with the MHSA module frozen, ViTPose obtains comparable performance to the fully finetuning setting.

**Task flexibility.** As the decoder is rather simple and lightweight in ViTPose, we can adopt multiple decoders without much extra cost to handle multiple pose estimation datasets by sharing the backbone encoder. We randomly sample instances from multiple training datasets for each iteration and feed them into the backbone and the decoders to estimate the heatmaps corresponding to each dataset.

### 3.4 The transferability of ViTPose

One common method to improve the performance of smaller models is to transfer the knowledge from larger ones, *i.e.*, knowledge distillation [17, 14]. Specifically, given a teacher network $T$ and student network $S$, a simple distillation method is to add an output distillation loss $L_{t \rightarrow s}^{od}$ to force the student network's output imitating the teacher network's output, *e.g.*,

$$L_{t \rightarrow s}^{od} = \text{MSE}(K_s, K_t), \tag{4}$$

where $K_s$ and $K_t$ are the outputs from the student and teacher network given the same input.

Apart from the above common practice, we explore a token-based distillation method to bridge the large and small models, which is complementary to the above method. Specifically, we randomly initialize an extra learnable knowledge token $t$ and append it to the visual tokens after the patch embedding layer of the teacher model. Then, we freeze the well-trained teacher model and only tune

the knowledge token for several epochs to gain the knowledge, *i.e.*,

$$t^* = \arg\min_t(\mathrm{MSE}(T(\{t; X\}), K_{gt}), \tag{5}$$

where $K_{gt}$ is the ground truth heatmaps, $X$ is the input images, $T(\{t; X\})$ denotes the predictions of the teacher, and $t^*$ represents the optimal token that minimizes the loss. After that, the knowledge token $t^*$ is frozen and concatenated with the visual tokens in the student network during training to transfer the knowledge from teacher to student networks. Thus, the loss of the student network is

$$L^{td}_{t\to s} = \mathrm{MSE}(S(\{t^*; X\}), K_{gt}), \quad or \quad L^{tod}_{t\to s} = \mathrm{MSE}(S(\{t^*; X\}), K_t) + \mathrm{MSE}(S(\{t^*; X\}), K_{gt}), \tag{6}$$

where $L^{td}_{t\to s}$ and $L^{tod}_{t\to s}$ represent the token distillation loss and the combination of output distillation loss and token distillation loss, respectively.

## 4 Experiments

### 4.1 Implementation details

ViTPose follows the common top-down setting for human pose estimation, *i.e.*, a detector is used to detect person instances and ViTPose is employed to estimate the keypoints of the detected instances. The detection results from SimpleBaseline [42] are utilized for evaluating ViTPose's performance on the MS COCO Keypoint val set. We use ViT-B, ViT-L, and ViT-H as backbones and denote the corresponding models as ViTPose-B, ViTPose-L, and ViTPose-H. The models are trained on 8 A100 GPUs based on the mmpose codebase [11]. The backbones are initialized with MAE [15] pre-trained weights. The default training setting in mmpose is utilized for training the ViTPose models, *i.e.*, we use the $256 \times 192$ input resolution and AdamW [33] optimizer with a learning rate of 5e-4. Udp [18] is used for post-processing. The models are trained for 210 epochs with a learning rate decay by 10 at the 170th and 200th epoch. We sweep the layer-wise learning rate decay [46] and stochastic drop path ratio for each model, and the optimal settings are provided in Table 1.

Table 1: Hyper-parameters for training ViTPose under the MS COCO only and multi-dataset settings. The hyper-parameters before and after the slash correspond to the MS COCO only setting and multi-dataset setting, respectively.

| Model | Batch Size | Learning rate | Weight decay | Layer wise decay | Drop path rate |
|-------|-----------|---------------|--------------|------------------|----------------|
| ViTPose-B | 512/1024 | 5e-4/1e-3 | 0.1 | 0.75 | 0.30 |
| ViTPose-L | 512/1024 | 5e-4/1e-3 | 0.1 | 0.80 | 0.50 |
| ViTPose-H | 512/1024 | 5e-4/1e-3 | 0.1 | 0.80 | 0.55 |
| ViTPose-G | 512/1024 | 5e-4/1e-3 | 0.1 | 0.85 | 0.55 |

### 4.2 Ablation study and analysis

Table 2: Ablation study of the structure simplicity of ViTPose on MS COCO val set.

| Backbone | ResNet-50 | | ResNet-152 | | ViTPose-B | | ViTPose-L | | ViTPose-H | |
|----------|-----------|--------|------------|--------|-----------|--------|-----------|--------|-----------|--------|
| Decoder | Classic | Simple | Classic | Simple | Classic | Simple | Classic | Simple | Classic | Simple |
| $AP$ | 71.8 | 53.1 | 73.5 | 55.3 | 75.8 | 75.5 | 78.3 | 78.2 | 79.1 | 78.9 |
| $AP_{50}$ | 89.8 | 86.9 | 90.5 | 87.9 | 90.7 | 90.6 | 91.4 | 91.4 | 91.7 | 91.6 |
| $AR$ | 77.3 | 62.0 | 79.0 | 63.8 | 81.1 | 80.9 | 83.5 | 83.4 | 84.1 | 84.0 |
| $AR_{50}$ | 93.7 | 92.1 | 94.3 | 92.9 | 94.6 | 94.6 | 95.3 | 95.3 | 95.4 | 95.4 |

**The structure simplicity and scalability.** We train ViTPose with the classic decoder and simple decoder as described in Sec. 3.1, respectively. We also train SimpleBaseline [42] with ResNet [16] as backbones using the two decoders for reference. Table 2 shows the results. It can be observed that using the simple decoder can lead to about 18 AP drops for both ResNet-50 and ResNet-152. However, ViTPose with vision transformer as backbones works well with the simple decoder with only marginal performance drops (*i.e.*, less than 0.3 AP) for ViT-B, ViT-L, and ViT-H. For the metrics $AP_{50}$ and $AR_{50}$, ViTPose obtains similar performance when using either of the two decoders, showing that the plain vision transformer has a strong representation ability and complex decoders are not necessary. It can also be concluded from the table that the performance of ViTPose improves consistently with the model size increasing, demonstrating the good scalability of ViTPose.

Table 3: The performance of ViTPose-B using different data for pre-training on MS COCO val set.

| Pre-training Dataset | Dataset Volume | $AP$ | $AP_{50}$ | $AP_{75}$ | $AR$ | $AR_{50}$ | $AR_{75}$ |
|---|---|---|---|---|---|---|---|
| ImageNet-1k | 1M | 75.8 | 90.7 | 83.2 | 81.1 | 94.6 | 87.7 |
| COCO (cropping) | 150K | 74.5 | 90.5 | 81.9 | 80.0 | 94.5 | 86.6 |
| COCO+AI Challenger (cropping) | 500K | 75.8 | 90.8 | 83.0 | 81.0 | 94.6 | 87.4 |
| COCO+AI Challenger (no cropping) | 300K | 75.8 | 90.5 | 83.0 | 81.0 | 94.5 | 87.4 |

**The influence of pre-training data.** To evaluate whether ImageNet-1K data are necessary for pose estimation tasks, we pre-train the backbone models using different datasets, *i.e.*, ImageNet-1k [12], MS COCO, and a combination of MS COCO [28] and AI Challenger [41], respectively. Since images in the ImageNet-1k dataset are iconic, we crop the person instances from the MS COCO and AI Challenger training set to form new training data for pre-training. The models are pre-trained for 1,600 epochs on the three datasets, respectively, and then finetuned on the MS COCO dataset with pose annotations for 210 epochs. The results are summarized in Table 3. It can be seen that with the combination of MS COCO and AI Challenger data for pre-training, ViTPose obtains comparable performance compared with using ImageNet-1k. It should be noted that the dataset volume is only half of the ImageNet-1k. It implies that pre-training on the data from downstream tasks has better data efficiency, validating ViTPose's flexibility in using pre-training data. Nevertheless, the AP decreases by 1.3 if only MS COCO data are used for pre-training. It may be caused by the limited volume of the MS COCO dataset, *i.e.*, the number of instances in MS COCO is three times less than the combination of MS COCO and AI Challenger. Besides, without the cropping operations, *i.e.*, directly using the images from MS COCO and AI Challenger for pre-training, ViTPose still obtains comparable performance compared with using cropping operations. This observation further validates the conclusion that the data from downstream tasks themselves can bring better data efficiency in the pre-training stage.

Table 4: The performance of ViTPose-B with different input resolutions on MS COCO val set.

| | 224x224 | 256x192 | 256x256 | 384x288 | 384x384 | 576x432 |
|---|---|---|---|---|---|---|
| $AP$ | 74.9 | 75.8 | 75.8 | 76.9 | 77.1 | 77.8 |
| $AR$ | 80.4 | 81.1 | 81.1 | 81.9 | 82.0 | 82.6 |

**The influence of input resolution.** To evaluate whether ViTPose can adapt well to different input resolutions, we train ViTPose with different input image sizes and give the results in Table 4. The performance of ViTPose-B improves with the increase of input resolution. It is also noted that the squared input does not bring much performance gains although it has larger resolutions, *e.g.*, $256 \times 256$ v.s. $256 \times 192$. The reason may be that the average aspect ratio of human instances in MS COCO is 4:3, and the squared input size does not fit the statistics well.

Table 5: The performance of ViTPose-B with 1/8 feature size on MS COCO val set. * means fp16 is used during training due to the limit of hardware memory. For the combination of full attention (Full) and window attention (Window), we follow ViTDet [25] and use full attention every 1/4 layers.

| Full | Window | Shift | Pool | Window Size | Training Memory (M) | GFLOPs | $AP$ | $AP_{50}$ | $AR$ | $AR_{50}$ |
|---|---|---|---|---|---|---|---|---|---|---|
| ✓ | | | | N/A | 36,141* | 76.59 | 77.4 | 91.0 | 82.4 | 94.9 |
| | ✓ | | | (8, 8) | 21,161 | 66.31 | 66.4 | 87.7 | 72.9 | 91.9 |
| | ✓ | ✓ | | (8, 8) | 21,161 | 66.31 | 76.4 | 90.9 | 81.6 | 94.5 |
| | ✓ | | ✓ | (8, 8) | 22,893 | 66.39 | 76.4 | 90.6 | 81.6 | 94.6 |
| | ✓ | ✓ | ✓ | (8, 8) | 22,893 | 66.39 | 76.8 | 90.8 | 81.9 | 94.8 |
| ✓ | ✓ | | | (8, 8) | 28,594 | 69.94 | 76.9 | 90.8 | 82.1 | 94.7 |
| | ✓ | ✓ | ✓ | (16, 12) | 26,778 | 68.46 | 77.1 | 91.0 | 82.2 | 94.8 |

**The influence of attention type.** As demonstrated in HRNet [36] and HRFormer [48], high-resolution feature maps are beneficial for pose estimation tasks. ViTPose can easily generate high-resolution features by varying the downsampling ratio of the patching embedding layer, *i.e.*, from 1/16 to 1/8. Besides, to alleviate the out-of-memory issue caused by the quadratic computational complexity of transformer layers, window attention with shift and pooling mechanism can be used as described in Sec. 3.3. The results are presented in Table 5. 'Shift' and 'Pool' denote the shift window and pooling window mechanisms, respectively. Directly using full attention with 1/8 feature size obtains the best 77.4 AP on the MS COCO val set while suffering from a large memory footprint even under the

mixed-precision training mode. Window attention can alleviate the memory issue while at the cost of performance drop due to lacking global context modeling, *e.g.*, from 77.4 AP to 66.4 AP. The shifted window and pooling window mechanism both promote cross-window information exchange for global context modeling and thus significantly improve the performance by 10 AP with less than 10% memory increase. When applying the two mechanisms together, *i.e.*, the 5th row, the performance further increases to 76.8 AP, which is comparable to the strategy proposed in ViTDet [25] that jointly uses full and window attention (the 6th row) but has much lower memory footprint, *i.e.*, 76.8 AP v.s. 76.9 AP and 22.9G memory v.s. 28.6G memory. Comparing the 5th and last row in Table 5, we also note that the performance can be further improved from 76.8 AP to 77.1 AP by enlarging the window size from $8 \times 8$ to $16 \times 12$, which also outperforms the joint full and window attention setting.

Table 6: The performance of ViTPose-B under the partially finetuning on MS COCO val set.

| FFN | MHSA | Memory (M) | GFLOPs | $AP$ | $AP_{50}$ | $AR$ | $AR_{50}$ |
|---|---|---|---|---|---|---|---|
| ✓ | ✓ | 14,090 | 17.1 | 75.8 | 90.7 | 81.1 | 94.6 |
| ✓ | | 11,052 | 10.9 | 75.1 | 90.5 | 80.3 | 94.4 |
| | ✓ | 10,941 | 6.2 | 72.8 | 89.8 | 78.3 | 93.8 |

**The influence of partially finetuning.** To assess whether vision transformers can adapt to the pose estimation task via partially finetuning, we finetune the ViTPose-B model under three settings, *i.e.*, fully finetuning, freezing the MHSA module, and freezing the FFN module. As shown in Table 6, with the MHSA module frozen, the performance drops a little compared with fully finetuning, *i.e.*, 75.1 AP v.s. 75.8 AP. The $AP_{50}$ metric is almost the same for the two settings. However, there is a significant drop by 3.0 AP when freezing the FFN module and only finetuning the MHSA module. This finding implies that the FFN module of vision transformers is more responsible for task-specific modeling. In contrast, the MHSA module is more task-agnostic, *e.g.*, modeling token relationships based on feature similarity no matter in the MIM pre-training tasks or specific pose estimation tasks.

Table 7: The performance of ViTPose-B under the multi-dataset training setting on MS COCO val set.

| COCO | AIC | MPII | $AP$ | $AP_{50}$ | $AR$ | $AR_{50}$ |
|---|---|---|---|---|---|---|
| ✓ | | | 75.8 | 90.7 | 81.1 | 94.6 |
| ✓ | ✓ | | 77.0 | 90.8 | 82.2 | 94.9 |
| ✓ | ✓ | ✓ | 77.1 | 90.8 | 82.2 | 94.7 |

**The influence of multi-dataset training.** Since the decoder in ViTPose is rather simple and lightweight, we can easily extend ViTPose to a multi-dataset joint training paradigm by using a shared backbone and individual decoder for each dataset. Specifically, we use MS COCO [28], AI Challenger [41], and MPII [3] datasets for multi-dataset training. The results on the MS COCO val set are listed in Table 7. The results on other datasets are available in the supplementary. Note that we directly use the models after multi-dataset training for evaluation without finetuning them on MS COCO further. It can be observed that the performance of ViTPose increases consistently from 75.8 AP to 77.1 AP by using all three datasets for training. Although the volume of MPII is much smaller compared to the combination of MS COCO and AI Challenger (40K v.s. 500K), using MPII for training still brings a 0.1 AP increase, indicating that ViTPose can well harness the diverse data in different datasets.

Table 8: The performance of transferability from ViTPose-L to ViTPose-B on MS COCO val set.

| Heatmap | Token | Teacher | Memory (M) | GFLOPs | $AP$ | $AP_{50}$ | $AR$ | $AR_{50}$ |
|---|---|---|---|---|---|---|---|---|
| - | - | - | 14,090 | 17.1 | 75.8 | 90.7 | 81.1 | 94.6 |
| | ✓ | ViTPose-L | 14,203 | 17.1 | 76.0 | 90.7 | 81.3 | 94.8 |
| ✓ | | ViTPose-L | 15,458 | 17.1 | 76.3 | 90.8 | 81.5 | 94.8 |
| ✓ | ✓ | ViTPose-L | 15,565 | 17.1 | 76.6 | 90.9 | 81.8 | 94.9 |

**The analysis of transferability.** To evaluate the transferability of ViTPose, we use both the classic output distillation and the proposed knowledge token distillation to transfer the knowledge from ViTPose-L to ViTPose-B. The results are available in Table 8. As can be seen, the token-based distillation brings 0.2 AP gain for ViTPose-B with marginal extra memory footprint, while the output distillation brings a 0.5 AP increase. The two distillation methods are complementary to each other, and using them together obtains 76.6 AP, validating the excellent transferability of ViTPose models.

## 4.3 Comparison with SOTA methods

Table 9: Comparison of ViTPose and SOTA methods on MS COCO val set. * denotes the models are trained under the multi-dataset setting.

| Model | Backbone | Params (M) | Speed (fps) | Input Resolution | Feature Resolution | COCO val AP | COCO val AR |
|---|---|---|---|---|---|---|---|
| SimpleBaseline [42] | ResNet-152 | 60 | 829 | 256x192 | 1/32 | 73.5 | 79.0 |
| HRNet [36] | HRNet-W32 | 29 | 916 | 256x192 | 1/4 | 74.4 | 78.9 |
| HRNet [36] | HRNet-W32 | 29 | 428 | 384x288 | 1/4 | 75.8 | 81.0 |
| HRNet [36] | HRNet-W48 | 64 | 649 | 256x192 | 1/4 | 75.1 | 80.4 |
| HRNet [36] | HRNet-W48 | 64 | 309 | 384x288 | 1/4 | 76.3 | 81.2 |
| UDP [18] | HRNet-W48 | 64 | 309 | 384x288 | 1/4 | 77.2 | 82.0 |
| TokenPose-L/D24 [27] | HRNet-W48 | 28 | 602 | 256x192 | 1/4 | 75.8 | 80.9 |
| TransPose-H/A6 [44] | HRNet-W48 | 18 | 309 | 256x192 | 1/4 | 75.8 | 80.8 |
| HRFormer-B [48] | HRFormer-B | 43 | 158 | 256x192 | 1/4 | 75.6 | 80.8 |
| HRFormer-B [48] | HRFormer-B | 43 | 78 | 384x288 | 1/4 | 77.2 | 82.0 |
| ViTPose-B | ViT-B | 86 | 944 | 256x192 | 1/16 | 75.8 | 81.1 |
| ViTPose-B* | ViT-B | 86 | 944 | 256x192 | 1/16 | 77.1 | 82.2 |
| ViTPose-L | ViT-L | 307 | 411 | 256x192 | 1/16 | 78.3 | 83.5 |
| ViTPose-L* | ViT-L | 307 | 411 | 256x192 | 1/16 | 78.7 | 83.8 |
| ViTPose-H | ViT-H | 632 | 241 | 256x192 | 1/16 | 79.1 | 84.1 |
| ViTPose-H* | ViT-H | 632 | 241 | 256x192 | 1/16 | 79.5 | 84.5 |

Based on the previous analysis, we use $256 \times 192$ input resolution with multi-dataset training for the pose estimation tasks and report the results on the MS COCO val and test-dev set as shown in Table 9 and Table 10. The speed of all methods is recorded on a single A100 GPU with a batch size of 64. It can be observed that although the model size of ViTPose is large, it obtains a better trade-off between throughput and accuracy, showing that the plain vision transformer has strong representation ability and is friendly to modern hardware. Besides, ViTPose performs well with much larger backbones. For example, ViTPose-L obtains much better performance than ViTPose-B, *i.e.*, 78.3 AP v.s 75.8 AP and 83.5 AR v.s. 81.1 AR on the val set. ViTPose-L has outperformed previous SOTA CNN and transformer models, including UPD and TokenPose, with a similar inference speed. Similar conclusions can be drawn by comparing the performance of ViTPose-H (15th row) and HRFormer-B (9th row), where ViTPose-H obtains better performance and faster inference speed, *i.e.*, 79.1 AP v.s. 75.6 AP and 241 fps v.s. 158 fps, with only MS COCO data for training. Besides, compared with the HRFormer [48], ViTPose has faster inference speed since its structure contains only one branch and operates on relative smaller feature resolution, *i.e.*, 1/16 compared with 1/4 used in HRFormer. With multi-dataset training, the performance of ViTPose models further increases, implying the good scalability and flexibility of ViTPose. This observation also demonstrates that with proper training and data, the plain vision transformer itself can model the relationships between different keypoints well and encode features of good linear separability for pose estimation tasks.

Table 10: Comparison with SOTA methods on MS COCO test-dev set. "+" means model ensemble. "†", "‡", and "*" denote the champions of the 2018, 2019, and 2020 COCO Keypoint Challenge.

| Method | Backbone | $AP$ | $AP_{50}$ | $AP_{75}$ | $AP_M$ | $AP_L$ | $AR$ |
|---|---|---|---|---|---|---|---|
| Baseline$^+$ [42] | ResNet-152 | 76.5 | 92.4 | 84.0 | 73.0 | 82.7 | 81.5 |
| HRNet [36] | HRNet-w48 | 77.0 | 92.7 | 84.5 | 73.4 | 83.1 | 82.0 |
| MSPN$^{++\dagger}$ [24] | 4xResNet-50 | 78.1 | 94.1 | 85.9 | 74.5 | 83.3 | 83.1 |
| DARK [49] | HRNet-w48 | 77.4 | 92.6 | 84.6 | 73.6 | 83.7 | 82.3 |
| RSN$^{++\ddagger}$ [8] | 4xRSN-50 | 79.2 | 94.4 | 87.1 | 76.1 | 83.8 | 84.1 |
| CCM$^+$ [50] | HRNet-w48 | 78.9 | 93.8 | 86.0 | 75.0 | 84.5 | 83.6 |
| UDP++$^{+*}$ [18] | HRNet-w48plus | 80.8 | 94.9 | 88.1 | 77.4 | 85.7 | 85.3 |
| ViTPose | ViTAE-G | 80.9 | 94.8 | 88.1 | 77.5 | 85.9 | 85.4 |
| **ViTPose$^+$** | **ViTAE-G** | **81.1** | **95.0** | **88.2** | **77.8** | **86.0** | **85.6** |

We then build a much stronger model ViTPose-G, *i.e.*, using the ViTAE-G [52] backbone, which has 1B parameters, larger input resolution ($576 \times 432$), and MS COCO and AI Challenger data for training, to further explore the ViTPose's performance limit. A more powerful detector from Bigdet [7] is also used to provide person detection results (68.5 AP on person class of COCO dataset).

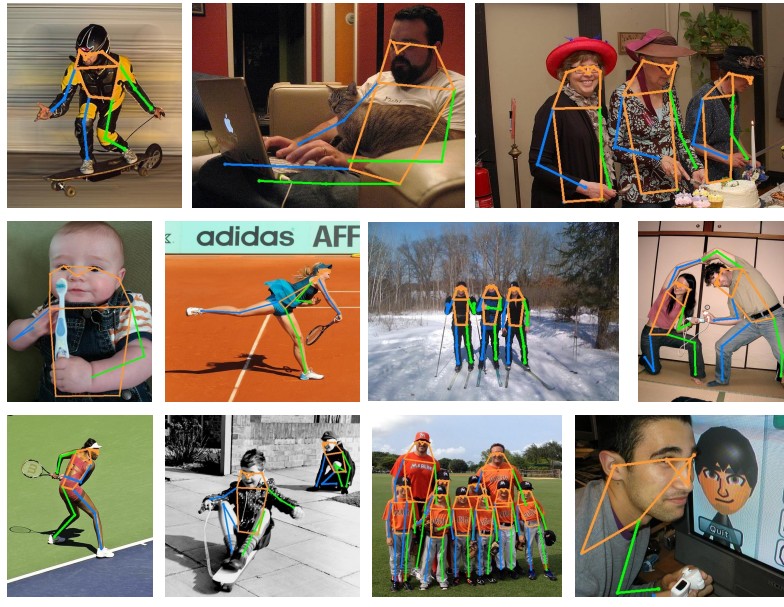

Figure 3: Visual pose estimation results of ViTPose on some test images from the MS COCO dataset.

As shown in Table 10, a single ViTPose model with the ViTAE-G backbone outperforms all previous SOTA methods on the MS COCO test-dev set at 80.9 AP, where the previous best method UDP++ ensembles 17 models and reaches 80.8 AP with a slightly better detector (68.6 AP on the person class of COCO dataset). After ensembling three models, ViTPose further achieves the best 81.1 AP.

### 4.4 Subjective results

We also visualize the pose estimation results of ViTPose on the MS COCO dataset. As shown in Figure 3, ViTPose can generate accurate pose estimation results on challenging cases with heavy occlusion, different postures, and different scales well, thanks to its good representation ability.

## 5 Limitation and Discussion

In this paper, we propose a simple yet effective vision transformer baseline for pose estimation, *i.e.*, ViTPose. Despite no elaborate designs in structure, ViTPose obtains SOTA performance on the MS COCO dataset. However, the potential of ViTPose is not fully explored with more advanced technologies, such as complex decoders or FPN structures, which may further improve the performance. Besides, although the ViTPose demonstrates exciting properties such as simplicity, scalability, flexibility, and transferability, more research efforts could be made, *e.g.*, exploring the prompt-based tuning to demonstrate the flexibility of ViTPose further. In addition, we believe ViTPose can also be applied to other pose estimation datasets, *e.g.*, animal pose estimation [47, 9, 45] and face keypoint detection [21, 6]. We leave them as the future work.

## 6 Conclusion

This paper presents ViTPose as the simple baseline for vision transformer-based human pose estimation. It demonstrates simplicity, scalability, flexibility, and transferability for the pose estimation tasks, which have been well justified through extensive experiments on the MS COCO dataset. A single ViTPose model with a big backbone ViTAE-G obtains the best 80.9 AP on the MS COCO test-dev set. We hope this work could provide useful insights to the community and inspire further study on exploring the potential of plain vision transformers in more computer vision tasks.

**Acknowledgement** Mr. Yufei Xu, Dr. Jing Zhang, and Mr. Qiming Zhang are supported by ARC FL-170100117 and IH-180100002.

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
