# A   Additional results of multi-dataset training

To evaluate the performance of ViTPose comprehensively, apart from the results on MS COCO val set, we also report the performance of ViTPose-B, ViTPose-L, ViTPose-H, and ViTPose-G on OCHuman [54] val and test set, MPII [3] val set, and AI Challenger [41] val set, respectively. Please note that the ViTPose variants are trained under the multi-dataset training setting and tested directly without further finetuning on the specific training dataset, to keep the whole pipeline as simple as possible.

*OCHuman val and test set.* To evaluate the performance of human pose estimation models on the human instances with heavy occlusion, we test the ViTPose variants and representative models on the OCHuman val and test set with ground truth bounding boxes. We do not adopt extra human detectors since not all human instances are annotated in the OCHuman datasets, where the human detector will cause a lot of "false positive" bounding boxes and can not reflect the true ability of pose estimation models. Specifically, the decoder head of ViTPose corresponding to the MS COCO dataset is used, as the keypoint definitions are the same in MS COCO and OCHuman datasets. The results are available in Table 11. Compared with previous state-of-the-art (SOTA) methods with complex structures, *e.g.*, MIPNet [20], ViTPose obtains over 10 AP increase on the OCHuman val set, although there is no special design to deal with occlusion in the network structure, implying the strong feature representation ability of ViTPose. It also should be noted that HRFormer [48] experiences large performance drops from MS COCO to OCHuman, and the small model beats the base model, *i.e.*, 53.1 AP v.s 50.4 AP on the OCHuman val set. Such phenomena imply that HRFormer may overfit to the MS COCO dataset, especially for lager-scale models, and need an extra finetuning stage to transfer from MS COCO to OCHuman. Besides, ViTPose significantly pushes forward the frontier of keypoint detection performance on both val and test set, *i.e.*, obtaining about 93 AP. Such results demonstrate that ViTPose can flexibly deal with challenging cases with heavy occlusion and obtain SOTA performance.

Table 11: Comparison of ViTPose and SOTA methods on OCHuman [54] val and test set with ground truth bounding boxes.

| Model | Backbone | Resolution | Val Set | | | | Test Set | | | |
|---|---|---|---|---|---|---|---|---|---|---|
| | | | $AP$ | $AP_{50}$ | $AR$ | $AR_{50}$ | $AP$ | $AP_{50}$ | $AR$ | $AR_{50}$ |
| SimpleBaseline [42] | ResNet-152 | 384x288 | 58.8 | 72.7 | 63.1 | 75.7 | 58.2 | 72.3 | 62.7 | 75.2 |
| HRNet [36] | HRNet-w32 | 384x288 | 60.9 | 76.0 | 65.1 | 78.2 | 60.6 | 74.8 | 64.7 | 77.6 |
| HRNet [36] | HRNet-w48 | 384x288 | 62.1 | 76.1 | 65.9 | 78.2 | 61.6 | 74.9 | 65.3 | 77.3 |
| MIPNet [20] | HRNet-w48 | 384x288 | 74.1 | 89.7 | 81.0 | - | - | - | - | - |
| HRFormer [48] | HRFormer-S | 384x288 | 53.1 | 73.1 | 59.6 | 76.9 | 52.8 | 72.8 | 59.1 | 76.6 |
| HRFormer [48] | HRFormer-B | 384x288 | 50.4 | 71.5 | 58.8 | 76.6 | 49.7 | 71.6 | 58.2 | 76.0 |
| ViTPose-B | ViT-B | 256x192 | 88.0 | 94.8 | 89.6 | 95.9 | 87.3 | 95.9 | 89.0 | 96.0 |
| ViTPose-L | ViT-L | 256x192 | 90.9 | 95.8 | 92.3 | 96.7 | 90.1 | 95.9 | 91.6 | 96.4 |
| ViTPose-H | ViT-H | 256x192 | 90.9 | 95.8 | 92.3 | 96.6 | 90.3 | 95.9 | 91.7 | 96.6 |
| ViTPose-G | ViTAE-G | 576x432 | 92.8 | 96.9 | 94.0 | 97.1 | 93.3 | 96.8 | 94.3 | 97.0 |

*MPII val set.* We evaluate the performance of ViTPose and representative models on the MPII val set with the ground truth bounding boxes. Following the default settings of MPII, we use PCKh as metric for performance evaluation. As demonstrated in Table 12, ViTPose variants obtain better performance on both single joint evaluation and average evaluation, *e.g.*, ViTPose-B, ViTPose-L, and ViTPose-H achieve 93.3, 94.0, and 94.1 average PCKh with smaller input resolutions (256x192 v.s. 256x256). With a larger input resolution and a larger backbone, *e.g.*, ViTPose-G with a ViTAE-G backbone and a 576x432 input resolution, the performance further increases to 94.3 PCKh, setting new SOTA on the MPII val set.

Table 12: Comparison of ViTPose and SOTA methods on MPII [3] val set with ground truth bounding boxes. PCKh is adopted as the evaluation metric.

| Model | Backbone | Resolution | Head | Shoulder | Elbow | Wrist | Hip | Knee | Ankle | Mean |
|---|---|---|---|---|---|---|---|---|---|---|
| SimpleBaseline [42] | ResNet-152 | 256x256 | 86.9 | 95.4 | 89.4 | 84.0 | 88.0 | 84.6 | 82.1 | 89.0 |
| HRNet [36] | HRNet-w32 | 256x256 | 96.9 | 85.9 | 90.5 | 85.9 | 89.1 | 86.1 | 82.5 | 90.0 |
| HRNet [36] | HRNet-w48 | 256x256 | 97.1 | 95.8 | 90.7 | 85.6 | 89.0 | 86.8 | 82.1 | 90.1 |
| CFA [35] | ResNet-101 | 384x384 | 95.9 | 95.4 | 91.0 | 86.9 | 89.8 | 87.6 | 83.9 | 90.1 |
| ASDA [5] | HRNet-w48 | 256x256 | 97.3 | 96.5 | 91.7 | 87.9 | 90.8 | 88.2 | 84.2 | 91.4 |
| TransPose-H-A6 [44] | HRNet-w48 | 256x256 | - | - | - | - | - | - | - | 92.3 |
| ViTPose-B | ViT-B | 256x192 | 97.5 | 97.4 | 93.7 | 90.5 | 92.3 | 91.5 | 88.1 | 93.3 |
| ViTPose-L | ViT-L | 256x192 | 97.8 | 97.6 | 94.3 | 91.2 | 93.0 | 92.5 | 89.8 | 94.0 |
| ViTPose-H | ViT-H | 256x192 | 97.7 | 97.6 | 94.4 | 91.5 | 93.2 | 92.6 | 90.3 | 94.1 |
| ViTPose-G | ViTAE-G | 576x432 | 98.0 | 97.6 | 94.5 | 91.9 | 92.9 | 93.0 | 90.2 | 94.3 |

*AI Challenger val set.* Similarly, we evaluate the performance of ViTPose on the AI Challenger val set with the corresponding decoder head. As summarized in Table 13, compared to representative CNN-based and transformer-based models, our ViTPose obtains better performance, *i.e.*, 35.4 AP from ViTPose-H v.s. 33.5 AP from HRNet-w48 and 34.4 AP from HRFromer base. ViTPose-G achieves the best 43.2 AP on the dataset with the stronger ViTAE-G backbone and a larger input resolution. However, the precision is still not high enough on the AI Challenger set, indicating that more efforts need to be made to further improve the performance.

Table 13: Comparison of ViTPose and SOTA methods on AI Challenger [41] val set with ground truth bounding boxes.

| Method | Backbone | Resolution | $AP$ | $AP_{50}$ | $AP_{75}$ | $AR$ | $AR_{50}$ |
|--------|----------|------------|------|-----------|-----------|------|-----------|
| SimpleBaseline [42] | ResNet-50 | 256x192 | 28.0 | 71.6 | 15.8 | 32.1 | 74.1 |
| SimpleBaseline [42] | ResNet-101 | 256x192 | 29.4 | 73.6 | 17.4 | 33.7 | 76.3 |
| SimpleBaseline [42] | ResNet-152 | 256x192 | 29.9 | 73.8 | 18.3 | 34.3 | 76.9 |
| HRNet [36] | HRNet-w32 | 256x192 | 32.3 | 76.2 | 21.9 | 36.6 | 78.9 |
| HRNet [36] | HRNet-w48 | 256x192 | 33.5 | 78.0 | 23.6 | 37.9 | 80.0 |
| HRFormer [48] | HRFomer-S | 256x192 | 31.6 | 75.9 | 20.9 | 35.8 | 78.0 |
| HRFormer [48] | HRFomer-B | 256x192 | 34.4 | 78.3 | 24.8 | 38.7 | 80.9 |
| ViTPose-B | ViT-B | 256x192 | 32.0 | 76.9 | 20.6 | 36.3 | 79.4 |
| ViTPose-L | ViT-L | 256x192 | 34.5 | 80.1 | 24.1 | 39.0 | 82.0 |
| ViTPose-H | ViT-H | 256x192 | 35.4 | 80.3 | 25.5 | 39.9 | 82.8 |
| ViTPose-G | ViTAE-G | 576x432 | 43.2 | 84.9 | 40.3 | 47.1 | 86.2 |

# B  Detailed dataset details.

*Dataset details.* We use MS COCO [28], AI Challenger [41], MPII [3], and CrowdPose [22] datasets for training and evaluation. OCHuman [54] dataset is only involved in the evaluation stage to measure the models' performance in dealing with occluded people. The MS COCO dataset contains 118K images and 150K human instances with at most 17 keypoint annotations each instance for training. The dataset is under the CC-BY-4.0 license. MPII dataset is under the BSD license and contains 15K images and 22K human instances for training. There are at most 16 human keypoints for each instance annotated in this dataset. AI Challenger is much bigger and contains over 200K training images and 350 human instances, with at most 14 keypoints for each instance annotated. OCHuman contains human instances with heavy occlusion and is just used for val and test set, which includes 4K images and 8K instances.

# C  Subjective results

We also provide some visual pose estimation results for subjective evaluation. We demonstrate the ViTPose results on AI Challenger (Figure 4), OCHuman (Figure 5), and MPII (Figure 6) datasets, respectively. Thanks to the strong representation ability and flexibility of ViTPose, it is good at dealing with challenging cases like occlusion, blur, appearance variance, irregular body postures, and *etc*.

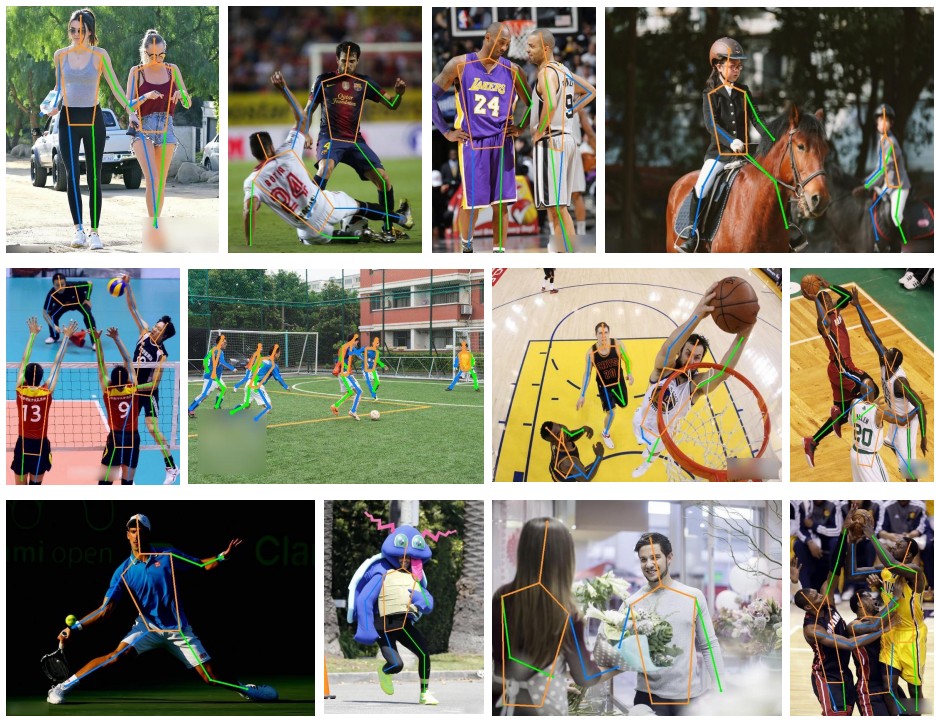

Figure 4: Visual pose estimation results of ViTPose on some test images from the AI Challenger dataset.

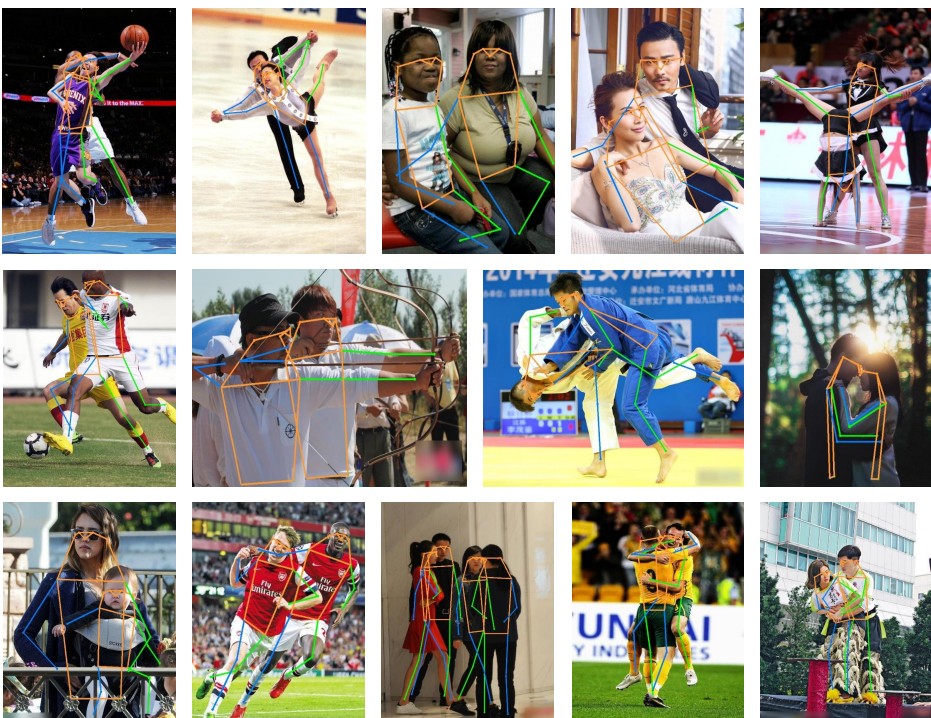

Figure 5: Visual pose estimation results of ViTPose on some test images from the OCHuman dataset.

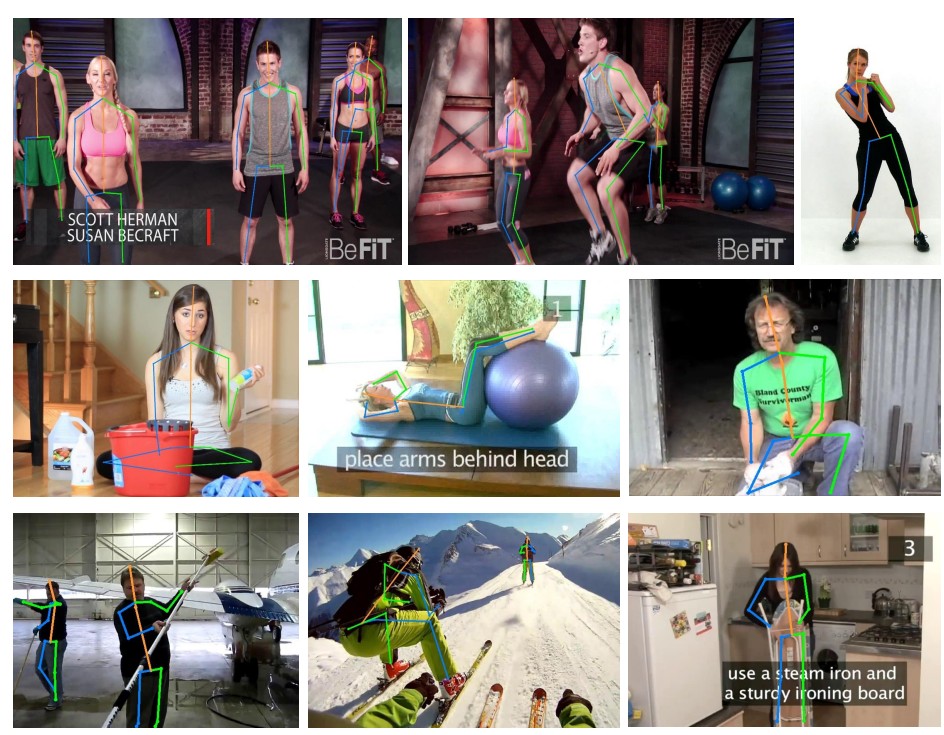

Figure 6: Visual pose estimation results of ViTPose on some test images from the MPII dataset.