# OpenReview forum: "ViTPose: Simple Vision Transformer Baselines for Human Pose Estimation"
_NeurIPS.cc/2022/Conference — NeurIPS 2022 Accept_

### Official Review · Reviewer_vNx3 · 2022-07-10

**Rating:** 6
**Confidence:** 4
**Soundness:** 3 good
**Presentation:** 3 good
**Contribution:** 2 fair

**Summary:**

This paper introduces a new baseline for pose estimation with vision transformer architecture. It obtains state-of-the-art performance on the MS COCO dataset even without the usage of elaborate structural designs or complex frameworks. Further, it has simplicity, scalability, flexibility and transferability.

**Questions:**

As listed in Weaknesses, the following questions need to be solved.

(1)	Introduce a smaller model for comparison with previous methods.

(2)	More explanations about this method and experimental results.


**Limitations:**

There exist some biased training data (e.g., gender, race, age) and personal privacy problems, as listed in this manuscript. This might be a general problem in the pose estimation community. These might be alleviated by masking face or using synthetic datasets.

**Strengths And Weaknesses:**

Pros:

(1)	A new baseline for pose estimation with vision transformer architecture is introduced.

(2)	SOTA results on MS COCO keypoint.

(3)	Comprehensive experiments show its simplicity, scalability, flexibility and transferability.

Cons:

This manuscript introduces a new baseline architecture, which achieves superior performance on the MS COCO keypoint dataset. I think it is a strong paper. However, there are still some questions.

In Table8, compared to HRFormer, ViTPose has more parameters but with a faster speed (e.g., ViTPose-B vs. HRFormer-B), which should be explained in detail. The authors are also encouraged to employ a smaller model (e.g., ViTPose-S or stack smaller layers) with comparable parameters to previous methods to show the superiority of this baseline. Besides, such a method in this manuscript should be a very simple baseline, and it is easy to think and implement, what is the reason for it surpassing the previous transformer-based method? It should be discussed.

---

> ### Author Response · Authors · 2022-08-02
> **Thanks and response to concerns**
>
> We sincerely thank you for the careful and thoughtful comments. Below we address the key concerns and promise will incorporate all feedback in the revised version.
>
> *1. In Table8, compared to HRFormer, ViTPose has more parameters but with a faster speed (e.g., ViTPose-B vs. HRFormer-B), which should be explained in detail.*
>
> **A1**: Thanks for your pointing out. We will add the discussion in the revised version. ViTPose-B employs plain vision transformers as encoders for feature extraction. The plain vision transformers consist of one single branch that operates on a relatively small feature resolution, i.e., 1/16 feature resolution. Thanks to that, ViTPose can benefit from the hardware-friendly operations in plain vision transformers, e.g., the attention and FFN modules of vision transformers are composed of parallel dense matrix multiplication operations, without the need to manipulate the memory with reshaping or resize operations frequently. HRFormer-B is also a fantastic work that adapts vision transformers for pose estimation. However, HRFormer employs a multi-branch structure with high-resolution branches, low-resolution branches, and interaction modules. The operation on high-resolution branches (1/4) is much more time-consuming for vision transformers, and thus other branches need to wait for the sync among the multiple branches. Besides, due to the interaction between multi-resolution features, HRFormer needs to frequently conduct reshaping and resize operations, which involves several memory manipulations. These memory operations break the pipeline of GPU execution and thus slow down the speed.
>
> *2. The authors are also encouraged to employ a smaller model (e.g., ViTPose-S or stack smaller layers) with comparable parameters to previous methods to show the superiority of this baseline.*
>
> **A2**: Thanks for your suggestion. We include a small model ViTPose-S in Table 8 for comparison. Generally, ViTPose-S with 24M parameters obtains **73.8 AP** with **1432 fps** on MS COCO. It is better than ResNet-50 (71.8 AP, 1351 fps) and ResNet-152 (73.5 AP, 829 fps) and comparable with the frontier transformer-based model HRFormer-S (73.8 AP, 269 fps). Compared with HRNet-w32 (74.6 AP, 916 fps), the performance of ViTPose-S is marginally weak, but it shows a much faster inference speed. The overview of performance and inference speed comparison is available in [link](https://ibb.co/n1RvgtZ). As demonstrated in the figure, ViTPose also sets the Parto Frontier in the small size models.
>
> *3. Besides, such a method in this manuscript should be a very simple baseline, and it is easy to think and implement. What is the reason for it surpassing the previous transformer-based method? It should be discussed.*
>
> **A3**: Thanks for your comments. It is a good question. Previous works like TokenPose, TransPose, and HRFormer obtain superior performance on pose estimation with vision transformers. However, most of them treat transformers as post-processors for feature enhancement, which do not fully explore the representation ability of transformers. HRFormer makes use of the strong representation ability of vision transformers but with task-favored multi-resolution designs. Recently, the community has demonstrated that plain vision transformers themselves could have strong representation abilities and perform well in image classification, object detection, segmentation, VQA, etc [1,2,3,4]. Such observation inspires us to rethink and explore **whether plain and simple vision transformers have strong representation ability in pose estimation tasks**. This direction has seldom been explored in previous studies. ViTPose makes a step to find the answer and surprisingly finds that plain vision transformers with good representation ability and proper training can solve pose estimation tasks well and demonstrates several good properties, including simplicity, scalability, flexibility, and transferability. **Owing to the strong representation ability of transformer blocks, the vision transformer itself can already encode features of good linear separability for pose estimation.** As a result, the decoder can be extremely simple (e.g., some bilinear upsample layers) and obtains a competitive performance. These findings can help us to further understand how to better adapt and design vision transformers for pose estimation tasks. We hope this study could serve as a starting point in adapting plain vision transformers for different vision tasks and exploring the interesting and beneficial properties of vision transformers.
>
> > [1] Bao, Hangbo, et al. "BEiT: BERT Pre-Training of Image Transformers." ICLR 2021.
>
> > [2] He, Kaiming, et al. "Masked autoencoders are scalable vision learners." CVPR 2022.
>
> > [3] Li, Yanghao, et al. "Exploring plain vision transformer backbones for object detection." ECCV 2022.
>
> > [4] Yu, Jiahui, et al. "Coca: Contrastive captioners are image-text foundation models." arXiv 2022.

---

> > ### Comment · Reviewer_vNx3 · 2022-08-09
> > **Response to Authors**
> >
> > Thanks for the authors’ response.
> >
> > The response fully solves my concerns. I will update my rating from 5 to 6.

---

> > > ### Author Response · Authors · 2022-08-09
> > > **Thanks for your effort and suggestions!**
> > >
> > > Thanks for your valuable comments and suggestions! We are encouraged by the resolution of your issue and appreciate your constructive comments to improve our work.

---

### Official Review · Reviewer_nUSy · 2022-07-10

**Rating:** 7
**Confidence:** 4
**Soundness:** 3 good
**Presentation:** 3 good
**Contribution:** 3 good

**Summary:**

This paper proposes a simple transformer based architecture for human pose estimation in images. The model is composed of a visual backbone (ViT) along with a simple decoder to obtain the heat maps providing the joint locations. The authors conduct multiple ablations to study the importance of various components (input resolution, number of parameters, ability to perform multi-task training etc). This result in a state of the art model on the COCO Keypoint Detection benchmark while maintaining a fast inference speed thanks to the ViT architecture which is modern hardware friendly.

**Questions:**

**Question/clarifications**

- What is the intuition behind the "token-based distillation"? Since it is trained to only work for a specific set of parameters, I am not sure to understand why adding it to a smaller model would suddenly distill knowledge from the big model to the small. In Table 7, the gains are not very clear (I am not super familiar with the type of variance one may encounter on the COCO keypoint detection benchmark, can the authors confirm whether or not those results are significant?). It would be great if the authors could comment on that point. If this is inspired by past work, it should be cited.

- What is layer-wise decay? How useful is that? Can you cite the paper for this?

**Minor typos**

- L151: double period at the end of the last sentence

**Limitations:**

Limitations and societal impact are adequately discussed in the paper.

**Strengths And Weaknesses:**

**Strengths**

- The paper is clearly written and well executed
- The method is simple to understand and performs well. Such strong baseline papers are important to the field as they can be the starting point for even better methods as it is easier to build on top of such approaches.
- I enjoyed reading the ablations study which is well conducted and properly shed lights on the important components of the approach

**Weaknesses**

- One could argue that there is not much novelty in the paper, as it is mainly a well executed strong baselines. However I still feel the paper in itself is a good enough contribution in itself.
- I am not sure to understand the motivation behind the learned token for transferability from large models to smaller models (see Questions below).

---

> ### Author Response · Authors · 2022-08-02
> **Thanks and response to concerns**
>
> We sincerely thank you for the careful and thoughtful comments. Below we address the key concerns and promise will incorporate all feedback in the revised version.
>
> *1. What is the intuition behind the "token-based distillation"? Since it is trained to only work for a specific set of parameters, I am not sure to understand why adding it to a smaller model would suddenly distill knowledge from the big model to the small. In Table 7, the gains are not very clear (I am not super familiar with the type of variance one may encounter on the COCO keypoint detection benchmark, can the authors confirm whether or not those results are significant?). It would be great if the authors could comment on that point. If this is inspired by past work, it should be cited.*
>
> **A1**: Thanks for your pointing out. Distillation of knowledge from the output is a common practice for both CNN and transformers. However, one major difference between CNN and transformer structure is the flexibility of inputs. For example, one can simply append extra tokens to the transformer’s inputs, as the attention and FFN modules of the transformer treat the inputs as 1D sequences. Previous works in classification [1] and NLP pretraining [2] tend to append an extra learnable token (cls toke or task token) to gather classification-sensitive information from the visual tokens and encode the task information to guide the network’s output. Inspired by these works, we wonder whether an extra token can serve for the knowledge distillation tasks by memorizing the knowledge from the larger models and guiding the smaller model via attention. To this end, we design a simple knowledge token-based distillation method and utilize the properties of vision transformers. Specifically, the knowledge token is trained to gather the pose knowledge from the larger models. After that, we freeze the knowledge toke and append it to the inputs of the smaller models to aid the smaller models’ training. Thanks to the flexibility of attention operations, the visual tokens can retrieve knowledge from the knowledge token in each transformer block and adapt its feature according to the knowledge, thus enabling small models to better focus on modeling the pose-related information. Different from the cls token and task token that is updated along with the models’ training and used for feature gathering, the knowledge token explored in the paper is frozen and expected to help the small models learn better feature representation for pose estimation tasks. Compared to the classic output-based distillation (+0.5 AP, +10\% training memory), this simple method brings about 40\% performance gain (+0.2 AP) with almost no extra training memory cost. Besides, it can also work with the classic output-based distillation to further improve the small models’ performance. It also should be noted that the exploration is just a starting point and baseline for exploring the flexibility of vision transformers for knowledge distillation. We hope it can provide some insights to the community and inspire new studies in exploring the transformers’ properties.
>
> > [1] Dosovitskiy, Alexey, et al. "An Image is Worth 16x16 Words: Transformers for Image Recognition at Scale." ICLR 2020.
>
> > [2] Kenton, et al. "BERT: Pre-training of Deep Bidirectional Transformers for Language Understanding." NAACL 2019.
>
> *2. What is layer-wise decay? How useful is that? Can you cite the paper for this?*
>
> **A2**: Layer-wise decay is a useful training technique for training both vision and NLP transformers [3,4,5,6]. It aims at setting different learning rates for different layers, i.e., $lr_i = lr * lrd^{(d-i)}$, where $i$ is the depth of the transformer layer, $d$ is the number of layers in the transformer model, and $lrd$ is the layer-wise learning rate decay rate. We disable the layer-wise learning rate decay for ViTPose-B to investigate the impact of this technique by setting each layer with the same learning rate. The performance of the model **drops from 75.8 AP to 74.4 AP**. It is because different layers of the vision transformer are responsible for encoding different information, i.e., the later layers are more task-specific than the former layers [7]. Thus, it is beneficial to tune the later layers faster than the former layers to adapt the vision transformer better and obtain better performance. We will provide the reference in the revised paper.
>
> > [3] Bao, Hangbo, et al. "BEiT: BERT Pre-Training of Image Transformers." ICLR 2021.
>
> > [4] He, Kaiming, et al. "Masked autoencoders are scalable vision learners."CVPR 2022.
>
> > [5] Xie, Zhenda, et al. "Simmim: A simple framework for masked image modeling." CVPR 2022.
>
> > [6] Yang, Zhilin, et al. "Xlnet: Generalized autoregressive pretraining for language understanding." Neurips 2019.
>
> > [7] Howard, Jeremy, and Sebastian Ruder. "Universal Language Model Fine-tuning for Text Classification." ACL 2018.

---

> > ### Comment · Reviewer_nUSy · 2022-08-09
> > **Thanks, final comment/question**
> >
> > Thank you for your answers and sorry for my late reply.
> >
> > I still think that the "distillation via the input" is not very well justified (it kind of make sense to try to distill a big model by forcing a small model to emulate its outputs), however even after reading the author's response it is still not clear why having those extra tokens coming from the big model would help to transfer knowledge (since those vectors are very close to the inputs -- they perform attention over input vectors, then its not clear how much more knowledge is store in those weights for a big model versus a small model) . Nonetheless if the results look to be significant (please confirm the AP improvements are not within noise), then maybe something is indeed happening that could deserve further investigation. I would advise the authors to highlight that this is only a starting exploration as I believe really making sure that this does distillation would deserve a more specific and detailed study.
> >
> > Thanks.

---

> > > ### Author Response · Authors · 2022-08-09
> > > **Thanks for your effort and suggestions!**
> > >
> > > Thanks for your valuable comments and suggestions! We will conduct the experiments with variant seeds and report more detailed results in the revised version. Besides, following your insightful suggestion, we will highlight that this is only a starting exploration in the revised version. We sincerely appreciate your constructive comments to improve our work.

---

### Official Review · Reviewer_ERVW · 2022-07-11

**Rating:** 6
**Confidence:** 5
**Soundness:** 3 good
**Presentation:** 3 good
**Contribution:** 3 good

**Summary:**

This paper applied the vision transformer to the task of single human pose estimation. The model uses plain and non-hierarchical vision transformers to extract features from the input image. Then a very simple decoder is added to regress heatmaps for human pose estimation. The method with a large version model achieved 80.9 AP on the COCO test-dev set. The authors also proposed a knowledge distillation method to transfer the knowledge from larger ViT models to smaller models.

**Questions:**

As in the weaknesses, there are some details to be clarified by the authors.

**Limitations:**

Yes. The authors described the limitation of not fully exploring more advanced technologies.

**Strengths And Weaknesses:**

Strengths
1) The paper is well-written and easy to read.
2) The model achieved state-of-the-art performance on the MSCOCO human pose estimation test-dev set.
3) The paper showed that with the powerful encoding ability of vision transformers, a very simple decoder (a bilinear upsample and one layer convolution) can achieve a very good result.
4) The authors also proposed a knowledge distillation method to transfer the knowledge from larger ViT models to smaller models, which improved the AP from 75.8 to 76.6 with little computation overhead.
5)  The authors did a lot of detailed ablation studies on the influences of partially fine-tuning, attention type, input resolution and pre-training data.
Weaknesses
1) Although the experiments of this paper are solid, the novelty of this paper is limited as this is an application of ViT to human pose estimations.
2) The claimed advantages of structural simplicity, model size scalability, training paradigm flexibility, and knowledge transferability can also be achieved by previous deep learning models. The vision transformer itself is not structurally simpler than CNN, stacking ViT without introducing other modules doesn't mean this model has a simpler structure compared to CNNs. Model size of other models can also be easily controlled by stacking. The training paradigm flexibility is very common in CNNs for humane pose estimation. Instead of using different decoders, particular output channels from different datasets can be masked during training due to the flexibility of heatmaps.
3) In lines 76-77, the authors claimed this model can be extended from a single pose dataset to multiple pose datasets. This can be confusing as in human pose estimation, multi-pose datasets normally refer to images with more than one pose to predict. It looks like the authors are describing extending the method to multi-dataset training.
4) The paper particular discussed "multi-task training". As far as I can tell, it's multi-dataset instead of multi-task training. The task is human pose estimation with different numbers of joints.

---

> ### Author Response · Authors · 2022-08-02
> **Thanks and response to concerns**
>
> We sincerely thank you for the careful and thoughtful comments. Below we address the key concerns and promise will incorporate all feedback in the revised version.
>
> *1. Although the experiments of this paper are solid, the novelty of this paper is limited as this is an application of ViT to human pose estimations.*
>
> **A1**: Thanks for your suggestion. As pointed out in paper Line *54* and Reviewer *nUSy* and *vNx3*, ViTPose aims to establish a strong and solid vision transformer baseline for human pose estimation tasks without elaborate structure design. It is based on a strong and solid motivation to *explore the strong representation ability* of plain vision transformers for pose estimation tasks. Most of the previous works treat vision transformers blocks as post-processers to model the relationships among different keypoints and refine the feature. Although they have shown promising results in pose estimation, they ignore the vision transformer’s, especially the plain vision transformer’s, strong representation power that the vision transformer itself can already encode features of good linear separability for pose estimation. ViTPose makes a step in this direction and supports the previous claim by reaching competitive performance with only one simple classifier, as shown in Table 1. Besides the good representation modeling ability, ViTPose also shows the community several valuable properties of vision transformers for pose estimation, including simplicity, scalability, flexibility, and transferability, which have not been explored previously. We hope these properties can provide the community with useful insights into adapting and utilizing vision transformers' strong representation ability for different tasks, including but not limited to pose estimation tasks. Having these baselines with simple vision transformers is beneficial to serve as starting points for future studies on pose estimation and other tasks.

---

> > ### Author Response · Authors · 2022-08-02
> > **Thanks and response to concerns (Part 2)**
> >
> > *2. The claimed advantages of simplicity, scalability, flexibility, and knowledge transferability can also be achieved by previous deep learning models. The vision transformer itself is not structurally simpler than CNN; stacking ViT without introducing other modules doesn't mean this model has a simpler structure compared to CNNs. Model size of other models can also be easily controlled by stacking. The training paradigm flexibility is very common in CNNs for humane pose estimation. Instead of using different decoders, particular output channels from different datasets can be masked during training due to the flexibility of heatmaps.*
> >
> > **A2**: A: Thanks for your suggestion. We will try to address these concerns from several aspects.
> >
> > 1. **The structure simplicity.** We think CNN is actually a good backbone and plays an important role in the development of deep learning for computer vision. We agree with you that the operation of convolution is simple and easy to complement in capturing local structures. To make the CNN models better at capturing global relationships, some fantastic module designs have been proposed, for example, SE attention [1], non-local attention [2], and ASPP [3]. These techniques have been proven beneficial in downstream vision tasks. Although they are not complex, it introduces extra structures like multi-branch structures. The vision transformer, instead, has a stronger representation ability even with an isotropic structure only because they could learn to acquire the inductive bias of locality and the ability to model long-range dependency from data. For example, ResNet is more adaptable to the classic decoder and experiences significant performance drops with the usage of simple bilinear upsample layers, while *ViTPose-B can obtain competitive performance even with the simple bilinear upsample layers*, thanks to the backbone’s strong representation ability, i.e., the vision transformer itself can already encode features of good linear separability for pose estimation. Thus, we think the usage of vision transformers can simplify the encoder and decoder’s design while obtaining competitive performance. Besides, the operations in vision transformers are composed of dense matrix multiplication operations, no matter in attention or FFN modules. Such operations are *hardware-friendly and computational simple* [4]. As demonstrated in Figure 1, it sets a new Parto Front for pose estimation tasks.
> >
> >     > [1] Hu, Jie, Li Shen, and Gang Sun. "Squeeze-and-excitation networks." CVPR 2018.
> >
> >     > [2] Wang, Xiaolong, et al. "Non-local neural networks." CVPR 2018.
> >
> >     > [3] Chen, Liang-Chieh, et al. "Deeplab: Semantic image segmentation with deep convolutional nets, atrous convolution, and fully connected crfs." TPAMI 2017.
> >
> >     > [4] Goto, Kazushige, et al. "Anatomy of high-performance matrix multiplication." ACM Transactions on Mathematical Software (TOMS) 34.3 (2008)
> >
> > 2. **The model scaling concerns.** Model scaling [5] is an important topic as it can enhance the model's representation ability and is a trend in future computer vision research for foundation models. However, as discussed in EfficientNet [6], the model’s performance diminishes by simply stacking the convolution layers, and it designs a compound scaling rule to scale the model sizes (about 400M parameters), which needs a heavy and careful model structure tuning process. Thus, it is not easy work to scale up the CNN models. Recently, transformer structures demonstrate a better scaling ability to over 3B parameters in both NLP and vision works, e.g., GPT-3 [7], Switch [8], and Swin-v2 [9] indicate a contiguous growth of representation ability in language or image classification tasks. To this end, we think it is necessary to evaluate ViTPose’s scaling ability and reveal whether the performance of large models will diminish in pose estimation tasks. Fortunately, we observe constant performance gains from ViTPose-B (88M) to ViTPose-G (1B). Such observation proves the scaling ability of vision transformers in pose estimation tasks, which has never been discussed before.
> >
> >     > [5] Bommasani, Rishi, et al. "On the opportunities and risks of foundation models." arXiv, 2021.
> >
> >     > [6] Tan, Mingxing, and Quoc Le. "Efficientnet: Rethinking model scaling for convolutional neural networks." ICML, 2019.
> >
> >     > [7] Floridi, Luciano, and Massimo Chiriatti. "GPT-3: Its nature, scope, limits, and consequences." Minds and Machines, 2020.
> >
> >     > [8] Fedus, William, Barret Zoph, and Noam Shazeer. "Switch Transformers: Scaling to Trillion Parameter Models with Simple and Efficient Sparsity." JMLR, 2022.
> >
> >     > [9] Liu, Ze, et al. "Swin transformer v2: Scaling up capacity and resolution." CVPR. 2022.

---

> > > ### Author Response · Authors · 2022-08-02
> > > **Thanks and response to concerns (Part 3)**
> > >
> > > *3. In lines 76-77, the authors claimed this model can be extended from a single pose dataset to multiple pose datasets. This can be confusing as in human pose estimation, multi-pose datasets normally refer to images with more than one pose to predict. It looks like the authors are describing extending the method to multi-dataset training.*
> > >
> > > **A3**: Sorry for the misunderstanding. We will make it clearer in the revised version. The multiple pose datasets mentioned in the paper represent joint training with different pose estimation datasets. During training, we randomly sample images from each dataset at each iteration and feed them into the network for feature extraction. Then, the extracted features are fed to dataset-specific heads for pose prediction and loss computation. Regarding whether there are multiple or single persons in the input image, we follow the common top-down pose estimation pipeline [10], using a person detector to detect each instance of a person in the input image, followed by the pose estimation algorithm to estimate the pose of each person.
> > >
> > > > [10] Xiao, Bin, Haiping Wu, and Yichen Wei. "Simple baselines for human pose estimation and tracking." ECCV. 2018.
> > >
> > > *4. The paper particularly discussed "multi-task training". As far as I can tell, it's multi-dataset instead of multi-task training. The task is human pose estimation with different numbers of joints.*
> > >
> > > **A4**: Sorry for the misunderstanding. We will make it clearer in the revised version. The multi-task training in the paper represents joint training with different pose estimation datasets. We denote each different pose estimation dataset as an individual pose estimation task since they have different posture distribution, keypoint annotations, and background scenes. We will change the term in the revised version. It should be noted that except for the different human pose estimation tasks discussed in the paper, we further extend the training paradigms to animal pose estimation (ap10k) [11], whole-body keypoint extraction (including face and foot) [12], and hand pose estimation tasks [13]. The results are demonstrated as follows. The results of ResNet and HRNet variants are taken from the MMPose website. Thanks to the strong representation ability of the vision transformer, ViTPose not only works well on human pose estimation across several tasks, but also performs well on the animal, whole body, and hand keypoint estimation.
> > >
> > > |            | COCO (AP) | AIC (AP) | MPII (PCKh) | CrowdPose (AP) | OChuman (AP) | AP10K (AP) | WholeBody (AP) | InterHand (AUC) |
> > > | :----------: | :---------: | :--------: | :-----------: | :--------------: | :------------: | :----------: | :--------------: | :---------------: |
> > > | ResNet-50  | 71.8      | \        | 88.2        | 63.7           | 54.6         | 68.1       | 52             | 85.1            |
> > > | ResNet-101 | 72.6      | 29.4     | 88.8        | 64.7           | 55.9         | 68.1       | 53.3           | \               |
> > > | ResNet-152 | 73.5      | \        | 88.9        | 65.6           | 57           | \          | 54.8           | \               |
> > > |            |           |          |             |                |              |            |                |                 |
> > > | HRNet-w32  | 74.6      | 32.3     | 90          | 67.5           | 59.1         | 72.2       | 55.3           | \               |
> > > | HRNet-w48  | 75.6      | 33.5     | 90.1        | \              | 61.1         | 73.1       | 57.9           | \               |
> > > |            |           |          |             |                |              |            |                |                 |
> > > | ViTPose-B  | 77.7      | 32.2     | 93.2        | 75.5           | 89.9         | 73.7       | 57.5           | 86.98           |
> > > | ViTPose-L  | 79.1      | 34.2     | 93.9        | 77.6           | 92.5         | 78.3       | 60.1           | 87.59           |
> > >
> > > > [11] Yu H, et al. AP-10K: A Benchmark for Animal Pose Estimation in the Wild, Neurips dataset track 2021.
> > >
> > > > [12] Jin, Sheng, et al. "Whole-body human pose estimation in the wild." ECCV, 2020.
> > >
> > > > [13] Moon, Gyeongsik, et al. "Interhand2. 6m: A dataset and baseline for 3d interacting hand pose estimation from a single rgb image." ECCV, 2020.

---

> > > > ### Comment · Reviewer_ERVW · 2022-08-09
> > > > **Thanks for the response**
> > > >
> > > > The authors have well addressed my concerns, especially on motivation and novelty.
> > > > The concern about training paradigm flexibility was not addressed. I still believe CNN has similar abilities as ViTPose. However, this is a minor concern. I hope the authors can carefully amend the paper as a few severe confusion is discussed above.
> > > > I will update my rating accordingly.

---

> > > > > ### Author Response · Authors · 2022-08-09
> > > > > **Thanks for your effort and suggestions!**
> > > > >
> > > > > Thanks for your valuable comments and suggestions! We are encouraged by the resolution of your major concerns and appreciate your constructive comments to improve our work. We promise that we will incorporate all feedback in the revised version and carefully amend the paper.

---

### Official Review · Reviewer_6Ljs · 2022-07-11

**Rating:** 6
**Confidence:** 5
**Soundness:** 3 good
**Presentation:** 3 good
**Contribution:** 3 good

**Summary:**

The article presents an experimental evaluation of the vision transformers for human pose estimation tasks. The experimental analysis focuses on transformer’s simplicity in model structure, scalability in model size, flexibility in training, and transferability of knowledge (distillation) from bigger models to smaller ones. The experimental evaluations are carried out on single MS COCO dataset by exploring various strategies such as attention types, image resolutions, pre-trained weights and finetuning. The largest models gives state-of-the-art accuracy on the MS COCO dataset and the authors have mentioned that the code and models will be released for other researchers. The evaluation on other datasets (CrowdPose, OCHuman, MPII, AI Challanger) is provided in supplementary material.

**Questions:**

1. Novelty of the approach

2. What is the difference between knowledge tokens and visual tokens? Which one is more important and why?

3. Model complexity should have included GFLOPS and trainable params on top of memory footprint.

4. What is the accuracy of COCO + AI challenger dataset  (Table 4) without using cropped person

5. In table 4:  Full+window training memory (28,594) is less than full only training memory (36,141). Any reason?

6. How the multi-task training carried out in Table 6?

**Limitations:**

The section 5 header says "Limitation" but the content does not discuss any limitations. However, it does mention about social impact of machine learning biases and personal privacy. It also mentions about the carbon footprint from data centers using this large-scale model training.

The paper should explore the training process and time taken to train such large models and how these models could be improved in near future.

**Strengths And Weaknesses:**

Strength:

The idea is good and is inspired by the lack of baseline evaluations of vision transformers for solving the pose estimation problem. The author has clarified that the paper does not claim algorithm superiority rather extensive evaluation on the MS COCO dataset to justify the benefits of vision transformers in solving pose estimation.

The paper has justified the usefulness of detailed experimental analysis by focusing on various aspect of vision transformer such as its structure, size, training process and knowledge distillation.

Experimental evaluation using well-known benchmarked human pose dataset of MS COCO is carried out. The performance of the proposed approach is compared to the some recent transformer and/or attention driven approaches.

Ablation study involving the benefit of pre-training data, influence of input resolution, impact of attention types, influence of partially finetuning and multi-task learning.

The article is well-written and easy to follow.

Weakness:

The proposed idea is an application of vision transformers to human pose estimation thus there is a question mark on the novelty.
The model’s computation complexity is presented as training memory. However, the other parameters such as GFLOPs and trainable parameters would have given a better reflection.

It is unclear how the proposed knowledge token is different form the visual tokens.

Although COCO + AI challenger dataset volume is half of ImageNet-1K (Table 2) but cropped person is used for training which is much cleaner dataset in comparison to ImageNet-1K. This could have influenced the performance gap. Any justification?

In table 4:  Full+window training memory (28,594) is less than full only training memory (36,141). Any reason?

In table 4: (16,12) window size is compared to (8,8) in the rest of the experiments except full. Any justification on selecting 16,12 why not 16,16?

How the multi-task training carried out in Table 6. Other than pose what other tasks that learned during multi-task learning.

---

> ### Author Response · Authors · 2022-08-02
> **Thanks and response to concerns**
>
> We sincerely thank you for the careful and thoughtful comments. Below we address the key concerns and promise will incorporate all feedback in the revised version.
>
> *1. Novelty of the approach.*
>
> **A1**: Thanks for your suggestion. As pointed out in paper Line *54* and Reviewer *nUSy* and *vNx3*, ViTPose aims to establish a strong and solid vision transformer baseline for human pose estimation tasks without elaborate structure design. Previous works have shown promising results in pose estimation with vision transformers. However, most of them treat vision transformers blocks as post-processers to model the relationships among different keypoints and refine the feature. They ignore the vision transformer's, especially the plain vision transformer's, strong representation ability that the vision transformer itself can already encode features of good linear separability for pose estimation. ViTPose makes a step in this direction and supports the previous claim by reaching competitive performance with only one simple classifier, as shown in Table 1. Besides the good representation modeling ability, ViTPose also shows the community several valuable properties of vision transformers for pose estimation, including simplicity, scalability, flexibility, and transferability, which have not been explored previously. We hope these properties can provide the community with useful insights into adapting and utilizing vision transformers' strong representation ability for different tasks, including but not limited to pose estimation tasks. Having these baselines with simple vision transformers is beneficial to serve as starting points for future studies on pose estimation and other tasks.
>
> *2. What is the difference between knowledge tokens and visual tokens? Which one is more important and why?*
>
> **A2**: Sorry for the misunderstanding. The knowledge token is trained using larger models and is *frozen* during training of smaller models. The visual tokens are directly generated from the input images via patch embedding layers, which are *continuously updated* during the training process. Apart from the difference in the optimization and generation mechanisms for these two types of tokens, their purposes differ in two aspects. 1) The knowledge token gathers useful information from the larger model and acts as a fixed “memory” variable in the small model. Intuitively, via the attention mechanism, the tokens in the small model could “read out” useful information from the knowledge token and learn to encode better feature representation. 2) The visual tokens are generated from the input images and are responsible for modeling the input's pose information. These two kinds of tokens can work together to improve the smaller models’ performance.

---

> > ### Author Response · Authors · 2022-08-02
> > **Thanks and response to concerns (Part 2)**
> >
> > *3. Model complexity should have included GFLOPS and trainable params on top of memory footprint.*
> >
> > **A3**: Thanks for your advice. Following your advice, we have updated the tables with memory footprint comparison by including GFLOPs and trainable parameters. Specifically,
> >
> > - Table 4
> >
> > | Full       | Window     | Shift      | Pool       | WindowSize | #Params (M) | GFLOPs | Memory (M) | $AP$    | $AP_{50}$ | $AR$    | $AR_{50}$ |
> > | :----------: | :----------: | :----------: | :----------: | :----------: | :-------: | :------: | :---------------: | :-----: | :-----: | :-----: | :-----: |
> > | $\checkmark$ |            |            |            | N/A        | 86      | 76.59  | 36,141           | 77.4  | 91.0  | 82.4  | 94.9  |
> > |            |            |            |            |            |         |        |                 |       |       |       |       |
> > |            | $\checkmark$ |            |            | (8, 8)     | 86      | 66.31  | 21,161           | 66.4  | 87.7  | 72.9  | 91.9  |
> > |            | $\checkmark$ | $\checkmark$ |            | (8, 8)     | 86      | 66.31  | 21,161           | 76.4  | 90.9  | 81.6  | 94.5  |
> > |            | $\checkmark$ |            | $\checkmark$ | (8, 8)     | 86      | 66.39  | 22,893           | 76.4  | 90.6  | 81.6  | 94.6  |
> > |            | $\checkmark$ | $\checkmark$ | $\checkmark$ | (8, 8)     | 86      | 66.39  | 22,893           | 76.8  | 90.8  | 81.9  | 94.8  |
> > | $\checkmark$ | $\checkmark$ |            |            | (8, 8)     | 86      | 69.94  | 28,594           | 76.9  | 90.8  | 82.1  | 94.7  |
> > |            |            |            |            |            |         |        |                 |       |       |       |       |
> > |            | $\checkmark$ | $\checkmark$ | $\checkmark$ | (16, 12)   | 86      | 68.46  | 26,778           | 77.1  | 91.0  | 82.2  | 94.8  |
> >
> > > We can see that the “no window attention” version has the most FLOPs compared with using window partitions for attention calculation. The number of trainable parameters is the same for all configurations since the attention calculation operation does not involve any parameters but different manners of matrix multiplication.
> >
> > - Table 5
> >
> > | FFN        | MHSA       | #Params (M) | GFLOPs | Memory (M) | $AP$   | $AP_{50}$ | $AR$   | $AR_{50}$ |
> > | :----------: | :----------: | :-------: | :------: | :------: | :----: | :-----: | :----: | :-----: |
> > | $\checkmark$ | $\checkmark$ | 86      | 17.1   | 14,090  | 75.8 | 90.7  | 81.1 | 94.6  |
> > | $\checkmark$ |            | 57      | 10.9   | 11,052  | 75.1 | 90.5  | 80.3 | 94.4  |
> > |            | $\checkmark$ | 23      | 6.2    | 10,941  | 72.8 | 89.8  | 78.3 | 93.8  |
> >
> > > We report the number of parameters and FLOPs of the trainable part and dismiss the frozen part. It can be observed that the FFN part contributes more parameters and FLOPs than the MHSA part.
> >
> > - Table 7
> >
> > | Token    | Heatmap      | Teacher   | #Params (M) | GFLOPs | Memory (M) | $AP$    | $AP_{50}$ | $AR$    | $AR_{50}$ |
> > | :----------: | :----------: | :---------: | :-------: | :------: | :------: | :-----: | :-----: | :-----: | :-----: |
> > | \          | \          | \         | 86      | 17.1   | 14,090  | 75.8  | 90.7  | 81.1  | 94.6  |
> > | $\checkmark$ |            | ViTPose-L | 86      | 17.1   | 14,203  | 76.0  | 90.7  | 81.3  | 94.8  |
> > |            | $\checkmark$ | ViTPose-L | 86      | 17.1   | 15,458  | 76.4  | 90.8  | 81.6  | 94.8  |
> > | $\checkmark$ | $\checkmark$ | ViTPose-L | 86      | 17.1   | 15,565  | 76.6  | 90.9  | 81.8  | 94.9  |
> >
> > > Since there is only one extra token introduced for the token-based distillation, the increase in extra FLOPs and parameters is negligible. However, although the teacher model is frozen and contributes zero to the FLOPs of the trainable part, it causes about a 10\% memory increase since we need to deploy the teacher model in GPU to provide online supervision given the input images.
> >
> > *4. What is the accuracy of COCO + AI challenger dataset (Table 4) without using cropped person*
> >
> > **A4**: Thanks for your suggestion. We conduct the experiment by directly using the images from the training set of the COCO and AI Challenger dataset without any cropping. We train the base models on the dataset for 1600 epochs following the same setting in Table 4 and then use the pre-trained weights to initialize the backbone models in ViTPose-B. The results are as follows. Using COCO + AIC without cropping also obtains competitive performance with using ImageNet-1k for pre-training, i.e., 75.8 AP, although the training data are much less than ImageNet-1k. It further validates the conclusion that pre-training on the data from downstream tasks has better data efficiency, and ViTPose is flexible in using the training data.
> >
> > | Dataset | Dataset Volume | $AP$ |
> > | :-------: | :--------------:| :----:|
> > | ImageNet-1k | 1M | 75.8 |
> > | COCO + AIC (cropping) | 500K | 75.8 |
> > | *COCO + AIC (no cropping)* | *300K* | *75.8* |

---

> > > ### Author Response · Authors · 2022-08-02
> > > **Thanks and response to concerns (Part 3)**
> > >
> > > *5. In table 4: Full+window training memory (28,594) is less than full only training memory (36,141). Any reason?*
> > >
> > > **A5**:  *‘Full+Window’* indicates that we use interleaved window-based and full attention in the model. Specifically, we insert full attention modules into the models every two window-based attention modules (one for per 1/4 depth position). We will briefly discuss the memory costs of window-based attention and full attention. Denote the resolution of the feature map as $H, W$, and the window size is $h, w$.
> > >
> > > - **For full attention modules**
> > >
> > > > Full attention module first projects the feature into $Q$, $K$, and $V$ with the same shape $(HW, C)$, where $C$ is the channel dimension. Then, it conducts the attention calculation as $Attn = (Q \times K^T)$, $Output = Attn \times V$. The matrix multiplication process of $Q$ and $K^T$ (with shape $(HW, C) \times (C, HW)$ consumes at least $(HW)^2C$ memory. Then, we get the attention matrix with the shape $(HW, HW)$. Then, we multiply the attention matrix with $V$, i.e., $(HW, HW) \times (HW, C)$. The process further consumes $(HW)^2C$ memory. Thus, the memory consumption of full attention is $2 (HW)^2C$.
> > >
> > > - Similarly, **for window-based attention**
> > >
> > > > Window-based attention modules first split the feature map into non-overlapping windows with size $(h, w)$. Then, we conduct the attention operation within each window. Thus, the memory consumption for each window is $2(hw)^2C$. The number of windows of the feature is $H/h \times W/w$. Thus, the total memory consumption of window-based attention is $2(hw)^2C \times H/h \times W/w$, which equals $2hwHWC$.
> > >
> > > Since the size of the window ($h, w$) is much smaller than the size of the feature ($H, W$), window-based attention consumes smaller memory than full attention operations. As observed in Table 4, the joint usage of full attention and window-based attention consumes less training memory than all full attention.
> > >
> > > *6. (16,12) window size is compared to (8,8) in the rest of the experiments except full. Any justification on selecting 16,12 why not 16,16?*
> > >
> > > **A6**: Thanks for your pointing out. We select (16, 12) instead of (16, 16) mainly due to the consideration of reducing extra computations. Firstly, the feature resolution of experiments in Table 4 is *32x24* given the typical input size *256x192* in the human pose estimation tasks. The window size of (16, 12) allows us to partition the feature into 4 non-overlapping windows exactly. In contrast, for the window size of *(16, 16)*, we need to pad the feature to *32x32* for window partition and then conduct the window-based attentions. Such a process of padding and then attention causes extra computational costs and memory consumptions, i.e., (16, 16) window partition causes an out-of-memory issue with full precision training and needs 26,780 training memory with fp16, while (16, 12) window partitions only need 26,778 training memory with full precision training.

---

> > > > ### Author Response · Authors · 2022-08-02
> > > > **Thanks and response to concerns (Part 4)**
> > > >
> > > > *7. How the multi-task training carried out in Table 6?*
> > > >
> > > > **A7**: Sorry for the misunderstanding. We will make it clearer in the revised version. The multi-task training represents joint training with different pose estimation datasets. We denote each different pose estimation dataset as an individual pose estimation task since they have different posture distribution, keypoint annotations, and background scenes. During training, we randomly sample images from each dataset at each iteration and feed them into the network for feature extraction. Then, the extracted features are fed to dataset-specific heads for pose prediction and loss computation. Except for the different human pose estimation tasks discussed in the paper, we further extend the training paradigms to animal pose estimation (ap10K [1]), whole-body keypoint extraction (including face and foot) [2], and hand pose estimation tasks [3]. The results are shown as follows. The results of ResNet and HRNet variants are taken from the MMPose repository. Thanks to the strong representation ability of the vision transformer, ViTPose not only works well on human pose estimation across several tasks but also performs well on the animal, whole body, and hand keypoint estimation tasks.
> > > >
> > > > |            | COCO (AP) | AIC (AP) | MPII (PCKh) | CrowdPose (AP) | OChuman (AP) | AP10K (AP) | WholeBody (AP) | InterHand (AUC) |
> > > > | :----------: | :---------: | :--------: | :-----------: | :--------------: | :------------: | :----------: | :--------------: | :---------------: |
> > > > | ResNet-50  | 71.8      | \        | 88.2        | 63.7           | 54.6         | 68.1       | 52             | 85.1            |
> > > > | ResNet-101 | 72.6      | 29.4     | 88.8        | 64.7           | 55.9         | 68.1       | 53.3           | \               |
> > > > | ResNet-152 | 73.5      | \        | 88.9        | 65.6           | 57           | \          | 54.8           | \               |
> > > > |            |           |          |             |                |              |            |                |                 |
> > > > | HRNet-w32  | 74.6      | 32.3     | 90          | 67.5           | 59.1         | 72.2       | 55.3           | \               |
> > > > | HRNet-w48  | 75.6      | 33.5     | 90.1        | \              | 61.1         | 73.1       | 57.9           | \               |
> > > > |            |           |          |             |                |              |            |                |                 |
> > > > | ViTPose-B  | 77.7      | 32.2     | 93.2        | 75.5           | 89.9         | 73.7       | 57.5           | 86.98           |
> > > > | ViTPose-L  | 79.1      | 34.2     | 93.9        | 77.6           | 92.5         | 78.3       | 60.1           | 87.59           |
> > > >
> > > > What’s more, ViTPose can also be extended to more vision tasks like object tracking, where the pose estimation and tracking pipeline can be jointly formulated using a single model for keypoint tracking tasks. We expect to see more research in this direction to build a foundation model for multiple different vision tasks.
> > > >
> > > > > [1] Yu H, et al. AP-10K: A Benchmark for Animal Pose Estimation in the Wild. Neurips dataset track 2021.
> > > >
> > > > > [2] Jin, Sheng, et al. "Whole-body human pose estimation in the wild." ECCV, 2020.
> > > >
> > > > > [3] Moon, Gyeongsik, et al. "Interhand2. 6m: A dataset and baseline for 3d interacting hand pose estimation from a single rgb image." ECCV, 2020.
> > > >
> > > > *8. The paper should explore the training process and time taken to train such large models.*
> > > >
> > > > **A8**: Thanks for pointing it out. The training of ViTPose-B takes about 4 hours with 8 A100 GPUs on the MS COCO dataset. Training ViTPose-L and ViTPose-H take 12h and 24h, respectively. The performance of these models can further be improved using several techniques like knowledge distillation [4] or features from multiple layers [5]. To accelerate the training process, some mechanisms like token pruning or sparse attention [6,7] can be further explored in the future. The adapter or prompt operations that have proven efficient in NLP works can also be explored further to reduce memory consumption [8,9]. As ViTPose just serves as a simple backbone and a starting point to demonstrate the good properties of plain vision transformers for pose estimation tasks, we believe there is so much that could be explored in future works.
> > > >
> > > > > [4] Wang, Wenhui, et al. "Minilm: Deep self-attention distillation for task-agnostic compression of pre-trained transformers." Neruips, 2020.
> > > >
> > > > > [5] Xie, Enze, et al. "SegFormer: Simple and efficient design for semantic segmentation with transformers." Neruips, 2021.
> > > >
> > > > > [6] Tang, Yehui, et al. "Patch slimming for efficient vision transformers." CVPR. 2022.
> > > >
> > > > > [7] Zaheer, Manzil, et al. "Big bird: Transformers for longer sequences." Neruips, 2020.
> > > >
> > > > > [8] Houlsby, Neil, et al. "Parameter-efficient transfer learning for NLP." ICML. PMLR, 2019.
> > > >
> > > > > [9] Lester, Brian, et al. "The Power of Scale for Parameter-Efficient Prompt Tuning." EMNLP, 2021.

---

### Author Response · Authors · 2022-08-02
**Thanks for the reviews**

We sincerely thank the reviewers for their thoughtful reviews. We are encouraged that the reviewers appreciate the solid and extensive experiments and ablation studies of ViTPose (Reviewer 6Ljs, ERVW, nUSy, vNx3), the SOTA performance of the proposed method (Reviewer 6Ljs, ERVW, nUSy, vNx3), the importance of the proposed baseline method for the pose estimation field (Reviewer 6Ljs, nUSy), the demonstration of various properties of vision transformers (Reviewer 6Ljs, nUSy, vNx3), and the well-written of the paper (Reviewer 6Ljs, ERVW, nUSy, vNx3).

We provide detailed responses to each reviewer, respectively, and promise we will incorporate all feedback in the revised version.

---

### Author Response · Authors · 2022-08-07
**Thanks again and looking forward to the discussion!**

We sincerely thank all the reviewers again for your thoughtful feedback and appreciate your efforts in reviewing the paper. We have endeavored to address the raised concerns regarding the difference between the knowledge and visual tokens and the computational complexity analysis from Reviewer 6Ljs, the updated results for training on multiple pose datasets with animal pose estimation, wholebody keypoint estimation, and hand keypoint estimation tasks in a joint learning pipeline from Reviewer 6Ljs and ERVW, the discussion of novelty and contribution of ViTPose from Reviewer 6Ljs and ERVW, the motivation of knowledge token design and the definition of layer-wise decay from Reviewer nUSy, and provide more analysis about the speed and performance of ViTPose to address the concerns raised by Reviewer vNx3. We are happy to discuss them with you in the openreview system if you feel that there still are some concerns/questions. We also welcome new suggestions/comments from you!

---

### Comment · Area_Chair_7UJf · 2022-08-09
**Please respond to the authors**

Dear reviewers,

The deadline is approaching, please respond to the authors and see whether they have addressed your raised issues.

Best,
Your AC

---

> ### Author Response · Authors · 2022-08-09
> **Thanks for the reminder**
>
> We sincerely thank the ACs for your kind reminder! We look forward to the reviewers' response and are willing to discuss these concerns with the reviewers further. Your thoughtful reviews help us a lot in improving the work.

---

### Meta-Review · Area_Chair_7UJf · 2022-08-25

**Recommendation:** Accept
**Confidence:** Certain

**Metareview:**

This submission received positive reviews. After rebuttal and discussions, all the reviewers feel positive about this submission with raised concerns addressed. After checking all the reviews and rebuttal, the AC stands on the reviewers' side and believe the current work is suitable for publication in this venue. The authors shall revise the manuscript according to the suggestions from the reviewers in the camera-ready submissions.

**Award:**

No

---

### Decision · Program_Chairs · 2022-09-14

Accept